# Analysis of Convolutions, Non-linearity and Depth in Graph Neural Networks using Neural Tangent Kernel

**Mahalakshmi Sabanayagam**                                        *sabanaya@cit.tum.de*
*School of Computation, Information and Technology*
*Technical University of Munich*

**Pascal Esser**                                        *esser@cit.tum.de*
*School of Computation, Information and Technology*
*Technical University of Munich*

**Debarghya Ghoshdastidar**                                        *ghoshdas@cit.tum.de*
*School of Computation, Information and Technology*
*Technical University of Munich*

**Reviewed on OpenReview:** *https://openreview.net/forum?id=xgYgDEof29*

## Abstract

The fundamental principle of Graph Neural Networks (GNNs) is to exploit the structural information of the data by aggregating the neighboring nodes using a 'graph convolution' in conjunction with a suitable choice for the network architecture, such as depth and activation functions. Therefore, understanding the influence of each of the design choice on the network performance is crucial. Convolutions based on graph Laplacian have emerged as the dominant choice with the symmetric normalization of the adjacency matrix as the most widely adopted one. However, some empirical studies show that row normalization of the adjacency matrix outperforms it in node classification. Despite the widespread use of GNNs, there is no rigorous theoretical study on the representation power of these convolutions, that could explain this behavior. Similarly, the empirical observation of the linear GNNs performance being on par with non-linear ReLU GNNs lacks rigorous theory.

In this work, we theoretically analyze the influence of different aspects of the GNN architecture using the *Graph Neural Tangent Kernel* in a semi-supervised node classification setting. Under the population *Degree Corrected Stochastic Block Model*, we prove that: (i) linear networks capture the class information as good as ReLU networks; (ii) row normalization preserves the underlying class structure better than other convolutions; (iii) performance degrades with network depth due to over-smoothing, but the loss in class information is the slowest in row normalization; (iv) skip connections retain the class information even at infinite depth, thereby eliminating over-smoothing. We finally validate our theoretical findings numerically and on real datasets such as *Cora* and *Citeseer*.

## 1 Introduction

With the advent of Graph Neural Networks (GNNs), there has been a tremendous progress in the development of computationally efficient state-of-the-art methods in various graph based tasks, including drug discovery, community detection and recommendation systems (Wieder et al., 2020; Fortunato & Hric, 2016; van den Berg et al., 2017). Many of these problems depend on the structural information of the data, represented by the graph, along with the features of the nodes. Because GNNs exploit this topological information encoded in the graph, it can learn better representation of the nodes or the entire graph than traditional deep learning techniques, thereby achieving state-of-the-art performances. In order to accomplish this, GNNs apply aggregation function to each node in a graph that combines the features of the neighboring nodes,

and its variants differ principally in the methods of aggregation. For instance, graph convolution networks use mean neighborhood aggregation through spectral approaches (Bruna et al., 2014; Defferrard et al., 2016; Kipf & Welling, 2017) or spatial approaches (Hamilton et al., 2017; Duvenaud et al., 2015; Xu et al., 2019), graph attention networks apply multi-head attention based aggregation (Velickovic et al., 2018) and graph recurrent networks employ complex computational module (Scarselli et al., 2008; Li et al., 2016). Of all the aggregation policies, the spectral graph Laplacian based approach is most widely used in practice, specifically the one proposed by Kipf & Welling (2017) owing to its simplicity and empirical success. In this work, we focus on such graph Laplacian based aggregations in Graph Convolution Networks (GCNs), which we refer to as *graph convolutions* or *diffusion operators.*

Kipf & Welling (2017) propose a GCN for node classification, a semi-supervised task, where the goal is to predict the label of a node using its feature and neighboring node information. They suggest symmetric normalization $\mathbf{S}_{sym} = \mathbf{D}^{-\frac{1}{2}}\mathbf{A}\mathbf{D}^{-\frac{1}{2}}$ as the graph convolution, where $\mathbf{A}$ and $\mathbf{D}$ are the adjacency and degree matrix of the graph, respectively. Ever since its introduction, $\mathbf{S}_{sym}$ remains the popular choice. However, subsequent works such as Wang et al. (2018); Wang & Leskovec (2020); Ragesh et al. (2021) explore row normalization $\mathbf{S}_{row} = \mathbf{D}^{-1}\mathbf{A}$ and particularly, Wang et al. (2018) observes that $\mathbf{S}_{row}$ outperforms $\mathbf{S}_{sym}$ for two-layered GCN empirically. Intrigued by this observation, and the fact that both $\mathbf{S}_{sym}$ and $\mathbf{S}_{row}$ are simply degree normalized adjacency matrices, we study the behavior over depth and observe that $\mathbf{S}_{row}$ performs better than $\mathbf{S}_{sym}$ in general, as illustrated in Figure 1 (Details of the experiment in Appendix C.1).

Furthermore, another striking observation from Figure 1 is that the performance of GCN without skip connections decreases considerably with depth for both $\mathbf{S}_{sym}$ and $\mathbf{S}_{row}$. This contradicts the conventional wisdom about standard neural networks which exhibit improvement in the performance as depth increases. Several works (Kipf & Welling, 2017; Chen et al., 2018b; Wu et al., 2019) observe this behavior empirically and attribute it to the over-smoothing effect from the repeated application of the diffusion operator, resulting in averaging out of the feature information to a degree where it becomes uninformative (Li et al., 2018; Oono & Suzuki, 2019; Esser et al., 2021). As a solution to this problem, Chen et al. (2020) and Kipf & Welling (2017) propose different forms of skip connections that overcome the smoothing effect and thus outperform the vanilla GCN. Extending it to the comparison of graph convolutions, Figure 1 shows $\mathbf{S}_{row}$ is preferable to

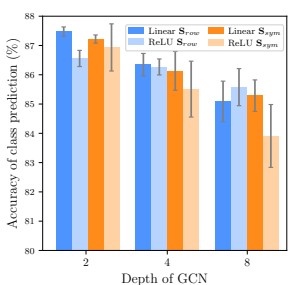

Figure 1: GCN performance on Cora dataset.

$\mathbf{S}_{sym}$ over depth in general for different GCNs. Naturally, we ask: *what characteristics of $\mathbf{S}_{row}$ enable better representation learning than $\mathbf{S}_{sym}$ in GCNs?* Another contrasting behavior to the standard deep networks is that *linear GCNs perform on par or even better than non-linear GCNs* as demonstrated in Wu et al. (2019). While standard neural networks with non-linear activations are proved to be universal function approximator, hence an essential component in a network, this behavior of GCNs is surprising.

Rigorous theoretical analysis is particularly challenging in GCNs compared to the standard neural networks because of the added complexity due to the graph convolution. Adding skip connections and non-linearity further increase the complexity of the analysis. To overcome these difficulties, we consider GCN in infinite width limit wherein the *Neural Tangent Kernel (NTK)* captures the network characteristics very well (Jacot et al., 2018). The infinite width assumption is not restrictive for our analysis as the NTK model shows same general trends as trained GCN. Moreover, NTK enables the analysis to be parameter-free and thus eliminate additional complexity induced, for example, by optimization. Through the lens of NTK, we study the impact of different graph convolutions under a random graph model: *Degree Corrected Stochastic Block Model (DC-SBM)* (Karrer & Newman, 2011). The node degree heterogeneity induced in DC-SBM allows us to analyze the effect of different types of normalization of the adjacency matrix, thus revealing the characteristic difference between $\mathbf{S}_{sym}$ and $\mathbf{S}_{row}$. Additionally, this model enables analysis of graphs that have *homophilic, heterophilic and core-periphery* structures. In this paper, we present a formal approach to analyze GCNs and, specifically, *the effect of activations, the representation power of different graph convolutions, the influence of depth and the role of skip connections.* This is a significant step toward understanding GCNs as it enables more informed network design choices like the convolution, depth and activations, as well as development of competitive methods based on grounded theoretical reasoning rather than heuristics.

**Contributions.** We provide a rigorous theoretical analysis of the discussed empirical observations in GCN under DC-SBM distribution using graph NTK, leading to the following contributions.

**(i)** In Sections 2–3, we present the NTK for GCN in infinite width limit in the node classification setting and our general framework of analysis, respectively.

**(ii)** In Section 4, we derive the NTK under DC-SBM and show that linear GCNs capture the class structure similar to ReLU GCN (or slightly better than ReLU) and, hence, linear GCN performs as good as ReLU GCNs. For convenience, we restrict the subsequent analysis to linear GCNs.

**(iii)** In Section 5, we show that for both homophilic and heterophilic graphs, row normalization preserves the class structure better, but is not useful in core-periphery models. We also derive that there is over-smoothing in vanilla GCN since the class separability decreases with depth.

**(iv)** In Section 6, we leverage the power of NTK to analyze different skip connections (Kipf & Welling, 2017; Chen et al., 2020). We derive the corresponding NTKs and show that skip connections retain class information even at infinite depth along with numerical validation.

Throughout the paper we illustrate the results numerically on planted models and validate the theoretical results on real dataset *Cora* in Section 7 and *Citeseer* in Appendix C.5, and conclude in Section 8 with the discussion on the impact of the results and related works. We provide all proofs, experimental details and more experiments in the appendix.

**Notations.** We represent matrix and vector by bold faced uppercase and lowercase letters, respectively, the matrix Hadamard (entry-wise) product by $\odot$ and the scalar product by $\langle .,. \rangle$. $\mathbf{M}^{\odot k}$ denotes Hadamard product of matrix $\mathbf{M}$ with itself repeated $k$ times. We use $\dot{\sigma}(.)$ for derivative of function $\sigma(.)$, $\mathbb{E}[.]$ for expectation, and $[d] = \{1, 2, \ldots, d\}$.

## 2 Neural Tangent Kernel for Graph Convolutional Network

Before going into a detailed analysis of graph convolutions we provide a brief background on *Neural Tangent Kernel* (NTK) and derive its formulation in the context of node level prediction using infinitely-wide GCNs. Jacot et al. (2018); Arora et al. (2019); Yang (2019) show that the behavior and generalization properties of randomly initialized wide neural networks trained by gradient descent with infinitesimally small learning rate is equivalent to a kernel machine. Furthermore, Jacot et al. (2018) also shows that the change in the kernel during training decreases as the network width increases, and hence, asymptotically, one can represent an infinitely wide neural network by a deterministic NTK, defined by the gradient of the network with respect to its parameters as

$$\mathbf{\Theta}(\mathbf{x}, \mathbf{x}') := \mathop{\mathbb{E}}_{\mathbf{W} \sim \mathcal{N}(\mathbf{0}, \mathbf{I})} \left[ \left\langle \frac{\partial F(\mathbf{W}, \mathbf{x})}{\partial \mathbf{W}}, \frac{\partial F(\mathbf{W}, \mathbf{x}')}{\partial \mathbf{W}} \right\rangle \right]. \tag{1}$$

Here $F(\mathbf{W}, \mathbf{x})$ represents the output of the network at data point $\mathbf{x}$ parameterized by $\mathbf{W}$ and the expectation is with respect to $\mathbf{W}$, where all the parameters of the network are randomly sampled from standard Gaussian distribution $\mathcal{N}(0, 1)$. Although the 'infinite width' assumption is too strong to model real (finite width) neural networks, and the absolute performance may not exactly match, the empirical trends of NTK match the corresponding network counterpart, allowing us to draw insightful conclusions. This trade-off is worth considering as this allows the analysis of over-parameterized neural networks without having to consider hyper-parameter tuning and training.

**Formal GCN Setup and Graph NTK.** We present the formal setup of GCN and derive the corresponding NTK, using which we analyze different graph convolutions, skip connections and activations. Given a graph with $n$ nodes and a set of node features $\{\mathbf{x}_i\}_{i=1}^n \subset \mathbb{R}^f$, we may assume without loss of generality that the set of observed labels $\{\mathbf{y}_i\}_{i=1}^m$ correspond to first $m$ nodes. We consider $K$ classes, thus $\mathbf{y}_i \in \{0, 1\}^K$ and the goal is to predict the $n - m$ unknown labels $\{\mathbf{y}_i\}_{i=m+1}^n$. We represent the observed labels of $m$ nodes as $\mathbf{Y} \in \{0, 1\}^{m \times K}$, and the node features as $\mathbf{X} \in \mathbb{R}^{n \times f}$ with the assumption that entire $\mathbf{X}$ is available during training. We define $\mathbf{S} \in \mathbb{R}^{n \times n}$ to be the graph convolution operator using the adjacency matrix $\mathbf{A}$ and the

degree matrix $\mathbf{D}$. The GCN of depth $d$ is given by

$$F_{\mathbf{W}}(\mathbf{X}, \mathbf{S}) := \sqrt{\frac{c_\sigma}{h_d}} \mathbf{S} \sigma \left( \ldots \sigma \left( \sqrt{\frac{c_\sigma}{h_1}} \mathbf{S} \sigma \left( \mathbf{S} \mathbf{X} \mathbf{W}_1 \right) \mathbf{W}_2 \right) \ldots \right) \mathbf{W}_{d+1} \tag{2}$$

where $\mathbf{W} := \{\mathbf{W}_i \in \mathbb{R}^{h_{i-1} \times h_i}\}_{i=1}^{d+1}$ is the set of learnable weight matrices with $h_0 = f$ and $h_{d+1} = K$, $h_i$ is the size of layer $i \in [d]$ and $\sigma : \mathbb{R} \to \mathbb{R}$ is the point-wise activation function where $\sigma(x) := x$ for linear and $\sigma(x) := \max(0, x)$ for ReLU activations. Note that linear $\sigma(x)$ is same as Simplified GCN (Wu et al., 2019). We initialize all the weights to be i.i.d standard Gaussian $\mathcal{N}(0, 1)$ and optimize it using gradient descent. We derive the NTK for the GCN in infinite width setting, that is, $h_1, \ldots, h_d \to \infty$. While this setup is similar to Kipf & Welling (2017), it is important to note that we consider linear output layer so that NTK remains constant during training (Liu et al., 2020) and a normalization $\sqrt{c_\sigma/h_i}$ for layer $i$ to ensure that the input norm is approximately preserved and $c_\sigma^{-1} = \underset{u \sim \mathcal{N}(0,1)}{\mathbb{E}} \left[ (\sigma(u))^2 \right]$ (similar to Du et al. (2019a)). The following theorem states the NTK between every pair of nodes, as a $n \times n$ matrix that can be computed at once.

**Theorem 1 (NTK for Vanilla GCN)** *For the vanilla GCN defined in* (2)*, the NTK $\boldsymbol{\Theta}$ at depth $d$ is*

$$\boldsymbol{\Theta}^{(d)} = \sum_{k=1}^{d+1} \mathbf{S} \underbrace{\left( \ldots \mathbf{S} \left( \mathbf{S} \left( \boldsymbol{\Sigma}_k \odot \dot{\mathbf{E}}_k \right) \mathbf{S}^T \odot \dot{\mathbf{E}}_{k+1} \right) \mathbf{S}^T \odot \ldots \odot \dot{\mathbf{E}}_d \right)}_{d+1-k \ terms} \mathbf{S}^T. \tag{3}$$

*Here $\boldsymbol{\Sigma}_k \in \mathbb{R}^{n \times n}$ is the co-variance between nodes of layer $k$, and is given by $\boldsymbol{\Sigma}_1 = \mathbf{S} \mathbf{X} \mathbf{X}^T \mathbf{S}^T$, $\boldsymbol{\Sigma}_k = \mathbf{S} \mathbf{E}_{k-1} \mathbf{S}^T$ with $\mathbf{E}_k = c_\sigma \underset{\mathbf{F} \sim \mathcal{N}(\mathbf{0}, \boldsymbol{\Sigma}_k)}{\mathbb{E}} \left[ \sigma(\mathbf{F}) \sigma(\mathbf{F})^T \right]$, $\dot{\mathbf{E}}_k = c_\sigma \underset{\mathbf{F} \sim \mathcal{N}(\mathbf{0}, \boldsymbol{\Sigma}_k)}{\mathbb{E}} \left[ \dot{\sigma}(\mathbf{F}) \dot{\sigma}(\mathbf{F})^T \right]$ and $\dot{\mathbf{E}}_{d+1} = \mathbf{1}_{n \times n}$.*

**Comparison to Du et al. (2019b).** While the NTK in (3) is similar to the graph NTK in Du et al. (2019b), the main difference is that NTK in our case is computed for all pairs of nodes in a graph as we focus on semi-supervised node classification, whereas Du et al. (2019b) considers supervised graph classification where input is many graphs and so the NTK is evaluated for all pairs of graphs. Moreover, the significant difference is in using the NTK to analytically characterize the influence of convolutions, non-linearity, depth and skip connections on the performance of GCN.

## 3   Theoretical Framework of our Analysis

In this section we discuss the general framework of our analysis that enables in substantiating different empirical observations in GCNs. We use the derived NTK in Theorem 1 for our analysis on various aspects of the GCN architecture and consider four different graph convolutions as defined in Definition 1 with Assumption 1 on the network.

**Definition 1** *Symmetric degree normalized $\mathbf{S}_{sym} := \mathbf{D}^{-\frac{1}{2}} \mathbf{A} \mathbf{D}^{-\frac{1}{2}}$, row normalized $\mathbf{S}_{row} := \mathbf{D}^{-1} \mathbf{A}$, column normalized $\mathbf{S}_{col} := \mathbf{A} \mathbf{D}^{-1}$ and unnormalized $\mathbf{S}_{adj} := \frac{1}{n} \mathbf{A}$ convolutions.*

**Assumption 1 (GCN with orthonormal features)** *GCN in* (2) *is said to have orthonormal features if $\mathbf{X} \mathbf{X}^T := \mathbf{I}_n$, where $\mathbf{I}_n$ is the identity matrix of size $n$.*

**Remark on Assumption 1.** The orthonormal features assumption eliminates the influence of the features and facilitates identification of the influence of different convolution operators clearly. Additionally, it helps in quantifying the exact interplay between the graph structure and different activation functions in the network. Nevertheless, the analysis including the features can be done using *Contextual Stochastic Block Model* (Deshpande et al., 2018) resulting in similar theoretical conclusions as detailed in Appendix B.9. Besides, the evaluation of our theoretical results without this assumption on real datasets is in Section 7 and Appendix C.5 that substantiate our findings.

While the NTK in (3) gives a precise characterization of the infinitely wide GCN, we can not directly draw conclusions about the convolution operators or activation functions without further assumptions on the input graph. Therefore, we consider a planted random graph model as described below.

**Random Graph Model.** We consider that the underlying graph is from the *Degree Corrected Stochastic Block Model (DC-SBM)* (Karrer & Newman, 2011) since it enables us to distinguish between $\mathbf{S}_{sym}$, $\mathbf{S}_{row}$, $\mathbf{S}_{col}$ and $\mathbf{S}_{adj}$ by allowing non-uniform degree distribution on the nodes. The model is defined as follows: Consider a set of $n$ nodes divided into $K$ latent classes (or communities), $\mathcal{C}_i \in [1, K]$. The DC-SBM model generates a random graph with $n$ nodes that has mutually independent edges with edge probabilities specified by the population adjacency matrix $\mathbf{M} = \mathbb{E}[\mathbf{A}] \in \mathbb{R}^{n \times n}$, where

$$\mathbf{M}_{ij} = \begin{cases} p\pi_i\pi_j & \text{if } \mathcal{C}_i = \mathcal{C}_j \\ q\pi_i\pi_j & \text{if } \mathcal{C}_i \neq \mathcal{C}_j \end{cases}$$

with the parameters $p, q \in [0, 1]$ governing the edge probabilities inside and outside classes, and the degree correction $\pi_i \in [0, 1] \, \forall \, i \in [n]$ with $\sum_i \pi_i = cn$ for a positive $c$ that controls the graph sparsity. The constant $c$ should be $\left[\frac{1}{\sqrt{n}}, 1\right]$ since the expected number of edges in this DC-SBM is $\mathcal{O}\left((cn)^2\right)$ and is bounded by $[n, n^2]$. Note that we deviate from the original condition $\sum_i \pi_i = K$ in Karrer & Newman (2011), to ensure that the analysis even holds for dense graphs. One can easily verify that the analysis holds for $\sum_i \pi_i = K$ as well. We denote $\boldsymbol{\pi} = (\pi_1, \ldots, \pi_n)$ for ease of representation. DC-SBM allows us to model different graphs: **Homophilic graphs:** $0 \leq q < p \leq 1$, **Heterophilic graphs:** $0 \leq p < q \leq 1$ and **Core-Periphery graphs:** $p = q$ (no assumption on class structure) and $\boldsymbol{\pi}$ encodes core and periphery. It is evident that the NTK is a complex quantity and computing its expectation is challenging given the dependency of terms from the degree normalization in $\mathbf{S}$, its powers $\mathbf{S}^i$ and $\mathbf{SS}^T$. To simplify our analysis, we make the following assumption on the DC-SBM,

**Assumption 2 (Population DC-SBM)** *The graph has a weighted adjacency* $\mathbf{A} = \mathbf{M}$.

**Remark on Assumption 2.** Assuming $\mathbf{A} = \mathbf{M}$ is equivalent to analyzing DC-SBM in expected setting and it further enables the computation of analytic expression for the population NTK instead of the expected NTK. Moreover, we empirically show that this analysis holds for random DC-SBM setting as well in Figure 5. Furthermore, this also implies addition of self loop with a probability $p$.

**Analysis Framework.** We analyze the observations of different GCNs by deriving the population NTK for each model and compare the preservation of class information in the kernel. Note that the true class information in the graph is determined by the blocks of the underlying DC-SBM – formally by $p$ and $q$ and independent of the degree correction $\boldsymbol{\pi}$. Consequently, we define the *class separability of the DC-SBM* as $r := \frac{p-q}{p+q}$. Hence, in order to capture the class information, the *kernel should ideally have a block structure that aligns with the one of the DC-SBM*. Therefore, we measure the *class separability of the kernel* as the average difference between in-class and out-of-class blocks. The best case is indeed when the class separability of the kernel is proportional (due to scale invariance of the kernel) to $p - q$ and independent of $\boldsymbol{\pi}$.

## 4   Linear Activation Captures Class Information as Good as ReLU Activation

While Kipf & Welling (2017) proposes ReLU GCNs, Wu et al. (2019) demonstrates that linear GCNs perform on par or even better than ReLU GCNs in a wide range of real world datasets, seemingly going against the notion that non-linearity is essential in neural networks. To understand this behavior, we derive the population NTK under DC-SBM for linear and ReLU GCNs, and compare the class separability of the kernels (average in-class and out-of-class block difference). Since our objective is in comparing linear and ReLU GCN, we consider homogeneous degree correction $\boldsymbol{\pi}$, that is, $\forall \, i, \pi_i := c$. In this case, population NTK for symmetric, row and column normalized adjacencies are equivalent, and unnormalized adjacency differ by a scaling that does not impact the block difference comparison. The following theorems state the population NTK for linear and ReLU GCNs of depth $d$ for normalized adjacency $\mathbf{S}$ and $K = 2$. The results hold for $K > 2$ as presented in Appendix B.3.5.

**Theorem 2 (Population NTK $\tilde{\Theta}$ for linear GCN)** *Let Assumption 1 and 2 hold, $\mathbb{1}[.]$ be indicator function, $K = 2$, $r := \frac{p-q}{p+q}$, $\delta_{ij} := (-1)^{\mathbb{1}[\mathcal{C}_i \neq \mathcal{C}_j]}$ and $\forall \, i, \pi_i := c$. Then $\forall \, i, j$, population NTK for linear GCN of*

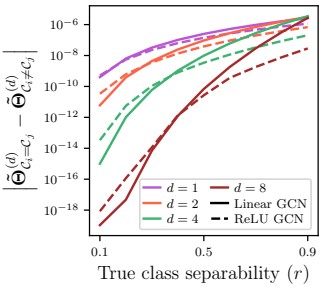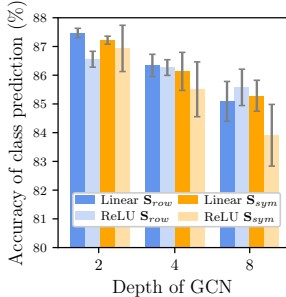

Figure 2: **Linear as good as ReLU activation. Left:** analytical plot of in-class and out-of-class block difference of the population NTK $\tilde{\mathbf{\Theta}}^{(d)}$ for a graph of size $n = 1000$, depths $d = \{1, 2, 4, 8\}$ and varying class separability $r$ of linear and ReLU GCNs (in log scale). **Right:** performance of trained linear and ReLU GCNs on *Cora* for $d = \{2, 4, 8\}$.

depth $d$, $\tilde{\mathbf{\Theta}}^{(d)}_{lin}$, is

$$\left(\tilde{\mathbf{\Theta}}^{(d)}_{lin}\right)_{ij} = \frac{d+1}{n}\left(1 + \delta_{ij} r^{2(d+1)}\right).$$

**Theorem 3 (Population NTK $\tilde{\mathbf{\Theta}}$ for ReLU GCN)** *Let assumptions of Theorem 2 hold and* $\kappa_0(x) := \frac{1}{\pi}\left(\pi - arccos\left(x\right)\right)$, $\kappa_1(x) := \frac{1}{\pi}\left(x\left(\pi - arccos\left(x\right)\right) + \sqrt{1-x^2}\right)$, $\Delta_1 := \frac{1-r^2}{1+r^2}$ *and* $\Delta_k := \frac{(1-r^2)+(1+r^2)\kappa_1(\Delta_{k-1})}{(1+r^2)+(1-r^2)\kappa_1(\Delta_{k-1})}$. *Furthermore,* $\Delta^n_k$ *and* $\Delta^d_k$ *denote the numerator and denominator of* $\Delta_k$, *respectively. Then* $\forall\ i, j$, *the population NTK for ReLU GCN of depth $d$,* $\tilde{\mathbf{\Theta}}^{(d)}_{ReLU}$, *is computed using* (3) *with*

$$(\mathbf{\Sigma}_k)_{ij} = \frac{1}{2^{k-1}n}\left(\mathbb{1}[\delta_{ij} = 1]\Delta^d_k + \mathbb{1}[\delta_{ij} = -1]\Delta^n_k\right)\prod_{k'=1}^{k-1}\Delta^d_{k'}$$

$$(\mathbf{E}_k)_{ij} = \frac{1}{2^{k-1}n}\left(\kappa_1\left(\Delta_k\right)\right)^{\mathbb{1}[\delta_{ij}=-1]}\prod_{k'=1}^{k}\Delta^d_{k'}\quad;\quad \left(\dot{\mathbf{E}}_k\right)_{ij} = \left(\kappa_0\left(\Delta_k\right)\right)^{\mathbb{1}[\delta_{ij}=-1]}.$$

**Comparison of Linear and ReLU GCNs.** The left of Figure 2 shows the analytic in-class and out-of-class block difference $\left|\tilde{\mathbf{\Theta}}^{(d)}_{\mathcal{C}_i=\mathcal{C}_j} - \tilde{\mathbf{\Theta}}^{(d)}_{\mathcal{C}_i\neq\mathcal{C}_j}\right|$ of the population NTKs of linear and ReLU GCNs with input graph size $n = 1000$ for different depths $d$ and class separability $r$. Given the class separability $r$ is large enough, theoretically *linear GCN preserves the class information as good as or slightly better than the ReLU GCN*. Particularly for $d = 1$, the difference is $\mathcal{O}\left(\frac{r^2}{n}\right)$ as shown in Appendix B.8. With depth, the difference prevails showing the effect of over-smoothing is stronger in ReLU than linear GCN, however larger depth proves to be detrimental for GCN as discussed in later sections. As a validation, we train linear and ReLU GCNs of depths $\{2, 4, 8\}$ on *Cora* dataset for both the popular convolutions $\mathbf{S}_{sym}$ and $\mathbf{S}_{row}$, and observe at par performance as shown in the right plot of Figure 2.

## 5   Convolution Operator $\mathbf{S}_{row}$ Preserves Class Information

In order to analyze the representation power of different graph convolutions $\mathbf{S}$, we derive the population NTKs under DC-SBM with non homogeneous degree correction $\boldsymbol{\pi}$ to distinguish the operators. We restrict our analysis to linear GCNs for convenience. In the following theorem, we state the population NTKs for graph convolutions $\mathbf{S}_{sym}$, $\mathbf{S}_{row}$, $\mathbf{S}_{col}$ and $\mathbf{S}_{adj}$ for $K = 2$ with Assumption 1 and 2. The result extends to $K > 2$ (Appendix B.3.5).

**Theorem 4 (Population NTKs $\tilde{\mathbf{\Theta}}$ and its class separability $\zeta$ for the four graph convolutions S)** *Let Assumption 1 and 2 hold,* $K = 2$ *and* $r := \frac{p-q}{p+q}$, $\delta_{ij} := (-1)^{\mathbb{1}[\mathcal{C}_i\neq\mathcal{C}_j]}$. $\boldsymbol{\pi}$ *is chosen such that*

$\sum_{i=1}^{n} \pi_i \mathbb{1}[\mathcal{C}_i = k] = \frac{cn}{K}$, $\sum_{i=1}^{n} \sqrt{\pi_i} \mathbb{1}[\mathcal{C}_i = k] = \tau \,\forall\, k$ and $\sum_{i=1}^{n} \pi_i^2 \mathbb{1}[\mathcal{C}_i = k] = \gamma \,\forall\, k$, where $\tau$ and $\gamma$ are constants. Then $\forall\, i, j$, population NTKs $\tilde{\mathbf{\Theta}}_{sym}$, $\tilde{\mathbf{\Theta}}_{row}$, $\tilde{\mathbf{\Theta}}_{col}$ and $\tilde{\mathbf{\Theta}}_{adj}$ and class separability of the population NTKs $\zeta_{sym}^{(d)}, \zeta_{row}^{(d)}, \zeta_{col}^{(d)}$ and $\zeta_{adj}^{(d)}$ of depth $d$ for $\mathbf{S} = \mathbf{S}_{sym}$, $\mathbf{S}_{row}$, $\mathbf{S}_{col}$ and $\mathbf{S}_{adj}$ respectively, are,

$$\left(\tilde{\mathbf{\Theta}}_{sym}^{(d)}\right)_{ij} = (d+1)\left(1 + \delta_{ij} r^{2d+2}\right) \frac{\sqrt{\pi_i \pi_j}}{cn} \qquad ; \zeta_{sym}^{(d)} = \frac{16\tau^2(d+1)}{n^2(cn)} r^{2d+2}$$

$$\left(\tilde{\mathbf{\Theta}}_{row}^{(d)}\right)_{ij} = (d+1)\left(1 + \delta_{ij} r^{2d+2}\right) \frac{2\gamma}{(cn)^2} \qquad ; \zeta_{row}^{(d)} = \frac{8\gamma(d+1)}{(cn)^2} r^{2d+2}$$

$$\left(\tilde{\mathbf{\Theta}}_{col}^{(d)}\right)_{ij} = (d+1)\left(1 + \delta_{ij} r^{2d+2}\right) \frac{n\pi_i \pi_j}{(cn)^2} \qquad ; \zeta_{col}^{(d)} = \frac{4(d+1)}{n} r^{2d+2}$$

$$\left(\tilde{\mathbf{\Theta}}_{adj}^{(d)}\right)_{ij} = (d+1)\pi_i \pi_j \frac{\gamma^{2^{d+1}-1}}{n^{2d+2}} \left( \mathbb{1}[\delta_{ij}=1] \sum_{l=0}^{2^d} \binom{2^{d+1}}{2l} p^{2^{d+1}-2l} + \right.$$

$$\left. \mathbb{1}[\delta_{ij}=-1] \sum_{l=0}^{2^d-1} \binom{2^{d+1}}{2l+1} p^{2^{d+1}-2l-1} q^{2l+1} \right) \qquad ; \zeta_{adj}^{(d)} = \frac{(d+1)c^2 \gamma^{2^{d+1}-1}}{n^{2d+2}} (p-q)^{2d+2}.$$

Note that the three assumptions on $\boldsymbol{\pi}$ are only to express the kernel in a simplified, easy to comprehend format. It is derived without the assumptions on $\boldsymbol{\pi}$ in Appendix B.3. Furthermore, the numerical validation of our result in Section 5.2 is without both these assumptions.

**Comparison of graph convolutions.** The population NTKs $\tilde{\mathbf{\Theta}}^{(d)}$ of depth $d$ in Theorem 4 describes the information that the kernel has after $d$ convolutions with $\mathbf{S}$. To classify the nodes perfectly, the kernels should retain the class information of the nodes according to the underlying DC-SBM. That is, the average in-class and out-of-class block difference of the population NTKs (class separability of the kernel) is proportional to $p - q$ and independent of $\boldsymbol{\pi}$. On this basis, only $\tilde{\mathbf{\Theta}}_{row}$ exhibits a block structure unaffected by the degree correction $\boldsymbol{\pi}$, and the average block difference is determined by $r^2$ and $d$, making $\mathbf{S}_{row}$ preferable over $\mathbf{S}_{sym}$, $\mathbf{S}_{adj}$ and $\mathbf{S}_{col}$. On the other hand, $\tilde{\mathbf{\Theta}}_{sym}$, $\tilde{\mathbf{\Theta}}_{col}$ and $\tilde{\mathbf{\Theta}}_{adj}$ are influenced by the degree correction $\boldsymbol{\pi}$ which obscures the class information especially with depth. Although $\tilde{\mathbf{\Theta}}_{sym}$ and $\tilde{\mathbf{\Theta}}_{col}$ seem similar, the influence of $\boldsymbol{\pi}$ for $\tilde{\mathbf{\Theta}}_{col}$ is $\mathcal{O}(\pi_i^2)$ which is stronger compared to $\mathcal{O}(\pi_i)$ for $\tilde{\mathbf{\Theta}}_{sym}$, making it undesirable over $\mathbf{S}_{sym}$. As a result, the preference order from the theory is $\tilde{\mathbf{\Theta}}_{row} \succ \tilde{\mathbf{\Theta}}_{sym} \succ \tilde{\mathbf{\Theta}}_{col} \succ \tilde{\mathbf{\Theta}}_{adj}$.

## 5.1 Impact of Depth in Vanilla GCN

Given that $r := \frac{p-q}{p+q} < 1$, Theorem 4 shows that the difference between in-class and out-of-class blocks decreases with depth monotonically which in turn leads to decrease in performance with depth, therefore explaining the observation in Figure 1. Corollary 1 characterizes the impact of depth as $d \to \infty$.

**Corollary 1 (Class separability of population NTK $\zeta^{(\infty)}$ as $d \to \infty$ )** *From Theorem 4, the class separability of population NTKs of the four different convolutions for fixed $n$ and as $d \to \infty$ converge to 0.*

Corollary 1 presents the class separability of the population NTKs for fixed $n$ and $d \to \infty$ for all the four convolutions $\mathbf{S}_{sym}$, $\mathbf{S}_{row}$, $\mathbf{S}_{col}$ and $\mathbf{S}_{adj}$, showing that the very deep GCN has zero class information. From this we also infer that, as $d \to \infty$ the population NTKs converge to a constant kernel, thus 0 average in-class and out-of-class block difference for all the convolutions. Therefore, *deeper GCNs have zero class information for any choice of convolution operator* $\mathbf{S}$. The class separability of population kernels at depth $d$ for $\mathbf{S}_{sym}$, $\mathbf{S}_{row}$ and $\mathbf{S}_{col}$ is $\mathcal{O}(\frac{dr^{2d}}{n})$ since $\tau$ and $\gamma$ are $\mathbf{O}(n)$. Therefore, it shows that *the class separation decreases at the exponential rate in $d$*. This explains the performance degradation of GCN with depth. To further understand the impact of depth, we plot the average in-class and out-of-class block difference for homophilic and heterophilic graphs using the theoretically derived population NTK $\tilde{\mathbf{\Theta}}^{(d)}$ for depths $[1, 10]$ and $n = 1000$ in a well separated DC-SBM (row 2, column 1 of Figure 3 and column 4 of Figure 4, respectively). It clearly

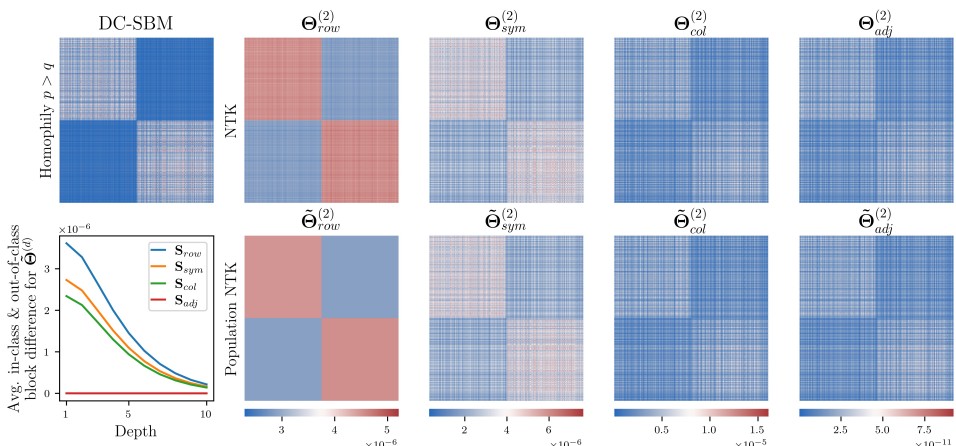

Figure 3: **Numerical validation of Theorem 4 using homophilic ($q < p$) DC-SBM** (Row 1, Column 1). Row 1, Columns 2–5 illustrate the exact NTKs of depth=2 and a graph of size $n = 1000$ sampled from the DC-SBM for $\mathbf{S}_{row}$, $\mathbf{S}_{sym}$, $\mathbf{S}_{col}$ and $\mathbf{S}_{adj}$. Row 2 shows the respective analytic population NTKs from Theorem 4. Row 2, column 1 shows the average gap between in-class and out-of-class blocks from theory, that is, average of $\left| \tilde{\mathbf{\Theta}}^{(d)}_{\mathcal{C}_i = \mathcal{C}_j} - \tilde{\mathbf{\Theta}}^{(d)}_{\mathcal{C}_i \neq \mathcal{C}_j} \right|$. This validates that $\mathbf{S}_{row}$ preserves class information better than other convolutions.

shows the exponential degradation of class separability with depth and the gap goes to 0 for large depths in all the four convolutions. Additionally, the gap in $\tilde{\mathbf{\Theta}}^{(d)}_{row}$ is the highest showing that the class information is better preserved, illustrating the strong representation power of $\mathbf{S}_{row}$. Therefore, *large depth is undesirable for all the convolutions in vanilla GCN and the theory suggests $\mathbf{S}_{row}$ as the best choice for shallow GCN.*

## 5.2 Numerical Validation for Random Graphs

Theorem 4 and Corollary 1 show that $\mathbf{S}_{row}$ has better representation power under Assumption 1 and 2, that is, for the linear GCN with orthonormal features and population DC-SBM. We validate this on homophilous and heterophilous random graphs of size $n = 1000$ with equal sized classes generated from DC-SBM. Figure 3 illustrates the results for depth=2 in the homophily case where the DC-SBM is presented in row 1 and column 1. We plot the NTKs of all the convolution operators computed from the sampled graph and the population NTKs as per the theory as heatmaps in rows 1 and 2, respectively. The heatmaps corresponding to the exact and the population NTKs clearly show that the class information for all the nodes is well preserved in $\mathbf{S}_{row}$ as there is a clear block structure than the other convolutions in which each node is diffused unequally due to the degree correction. Among $\mathbf{S}_{sym}, \mathbf{S}_{col}$ and $\mathbf{S}_{adj}$, $\mathbf{S}_{sym}$ retains the class structure better and $\mathbf{S}_{adj}$ has very small values (see the colorbar scale) and no clear structure. Thus, exhibiting the theoretically derived preference order. We plot both the exact and the populations NTKs to show that the population NTKs are a good representative of the exact NTKs especially for large graphs. We show this by plotting the norm of relative kernel difference, $\| \frac{\tilde{\mathbf{\Theta}}^{(d)} - \mathbf{\Theta}^{(d)}}{\tilde{\mathbf{\Theta}}^{(d)}} \|_2$, with graph size $n$ for $d = 2$ in Figure 5. Figure 4 shows the analogous result for heterophily DC-SBM. The experimental details are provided in the Appendix C.3.

## 5.3 $\mathbf{S}_{sym}$ Maybe Preferred Over $\mathbf{S}_{row}$ in Core-Periphery Networks (No Class Structure)

While we showed that the graph convolution $\mathbf{S}_{row}$ preserves the underlying class structure, it is natural to wonder about the random graphs that have no communities ($p = q$). One such case is graphs with core-periphery structure where the graph has core nodes that are highly interconnected and periphery nodes that are sparsely connected to the core and other periphery nodes. Such a graph can be modeled using only the degree correction $\boldsymbol{\pi}$ such that $\pi_j \ll \pi_i \; \forall j \in periphery, i \in core$ (similar to Jia & Benson (2019)). Extending Theorem 4, we derive the following Corollary 2 and show that the convolution $\mathbf{S}_{sym}$ contains the graph information while $\mathbf{S}_{row}$ is a constant kernel.

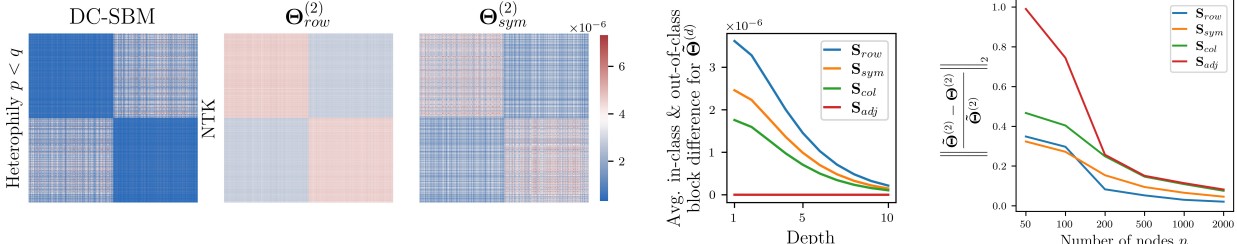

Figure 4: **Numerical validation of Theorem 4 using heterophilic ($p < q$) DC-SBM** (Column 1). Columns 2–3 illustrate the exact NTKs of depth=2 and a graph of size $n = 1000$ sampled from the DC-SBM for $\mathbf{S}_{row}$ and $\mathbf{S}_{sym}$. Column 4 shows the average gap between in-class and out-of-class blocks from theory.

Figure 5: Norm of the relative kernel difference $\|\frac{\tilde{\Theta}^{(2)} - \Theta^{(2)}}{\tilde{\Theta}^{(2)}}\|_2$ for depth $d = 2$ with graph size $n$.

**Corollary 2 (Population NTKs $\tilde{\Theta}$ for $p = q$)** *Let Assumption 1 and 2 hold, $K = 2$ and $p = q$. Furthermore, $\boldsymbol{\pi}$ is chosen such that $\sum_{i \in core} \pi_i^2 = \lambda$ and $\sum_{i \in periphery} \pi_i^2 = \mu$. Then $\forall$ $i$ and $j$, the population NTKs $\tilde{\Theta}_{sym}$ and $\tilde{\Theta}_{row}$ of depth $d$ for $\mathbf{S} = \mathbf{S}_{sym}$ and $\mathbf{S}_{row}$, respectively, are,*

$$\left(\tilde{\Theta}_{sym}^{(d)}\right)_{ij} = (d+1)\frac{\sqrt{\pi_i \pi_j}}{cn} \quad and \quad \left(\tilde{\Theta}_{row}^{(d)}\right)_{ij} = (d+1)\frac{\lambda + \mu}{(cn)^2}.$$

From Corollary 2, it is evident that *the $\mathbf{S}_{sym}$ has the graph information and hence could be preferred when there is no community structure.* We validate it experimentally and discuss the results in Figure 18 of Appendix C.3. While $\mathbf{S}_{row}$ results in a constant kernel for core-periphery without community structure, it is important to note that when there exists a community structure and each community has core-periphery nodes, then $\mathbf{S}_{row}$ is still preferable over $\mathbf{S}_{sym}$ as it is simply a special case of homophilic networks. This is demonstrated in Figure 19 of Appendix C.3.

## 6 Skip Connections Retain Class Information Even at Infinite Depth

Skip connection is the most common way to overcome the performance degradation with depth in GCNs, but little is known about the effectiveness of different skip connections and their interplay with the convolutions. While our focus is to understand the interplay with convolutions, we also include the impact of convolving with and without the feature information. Hence, we consider the following two variants: Skip-PC (pre-convolution), where the skip is added to the features before applying convolution (Kipf & Welling, 2017); and Skip-$\alpha$, which gives importance to the features by adding it to each layer without convolving with $\mathbf{S}$ (Chen et al., 2020). To facilitate skip connections, we need to enforce constant layer size, that is, $h_i = h_{i-1}$. Therefore, we transform the input layer using a random matrix $\mathbf{W}$ to $\mathbf{H}_0 := \mathbf{X}\mathbf{W}$ of size $n \times h$ where $\mathbf{W}_{ij} \sim \mathcal{N}(0, 1)$ and $h$ is the hidden layer size. Let $\mathbf{H}_i$ be the output of layer $i$.

**Definition 2 (Skip-PC)** *In a Skip-PC (pre-convolution) network, the transformed input $\mathbf{H}_0$ is added to the hidden layers before applying the graph convolution $\mathbf{S}$, that is, $\forall i \in [d], \mathbf{H}_i := \sqrt{\frac{c_\sigma}{h}}\mathbf{S}\left(\mathbf{H}_{i-1} + \sigma_s\left(\mathbf{H}_0\right)\right)\mathbf{W}_i$, where $\sigma_s(.)$ can be linear or ReLU.*

Skip-PC definition deviates from Kipf & Welling (2017) in the fact that we skip to the input layer instead of the previous layer. The following defines the skip connection similar to Chen et al. (2020).

**Definition 3 (Skip-$\alpha$)** *Given an interpolation coefficient $\alpha \in (0, 1)$, a Skip-$\alpha$ network is defined such that the transformed input $\mathbf{H}_0$ and the hidden layer are interpolated linearly, that is, $\mathbf{H}_i := \sqrt{\frac{c_\sigma}{h}}\left((1 - \alpha)\mathbf{S}\mathbf{H}_{i-1} + \alpha\sigma_s\left(\mathbf{H}_0\right)\right)\mathbf{W}_i$ $\forall i \in [d]$, where $\sigma_s(.)$ can be linear or ReLU.*

## 6.1 NTK for GCN with Skip Connections

We derive NTKs for the skip connections – Skip-PC and Skip-$\alpha$ by considering the hidden layers width $h \to \infty$. Both the NTKs maintain the form presented in Theorem 1 with the following changes to the co-variance matrices. Let $\tilde{\mathbf{E}}_0 = \underset{\mathbf{F} \sim \mathcal{N}(\mathbf{0}, \boldsymbol{\Sigma}_0)}{\mathbb{E}} \left[ \sigma_s(\mathbf{F}) \sigma_s(\mathbf{F})^T \right]$.

**Corollary 3 (NTK for Skip-PC)** *The NTK for an infinitely wide Skip-PC network is as presented in Theorem 1 where $\mathbf{E}_k$ is defined as in the theorem, but $\boldsymbol{\Sigma}_k$ is defined as*

$$\boldsymbol{\Sigma}_0 = \mathbf{X}\mathbf{X}^T, \qquad \boldsymbol{\Sigma}_1 = \mathbf{S}\tilde{\mathbf{E}}_0\mathbf{S}^T \qquad and \qquad \boldsymbol{\Sigma}_k = \mathbf{S}\mathbf{E}_{k-1}\mathbf{S}^T + \boldsymbol{\Sigma}_1.$$

**Corollary 4 (NTK for Skip-$\alpha$)** *The NTK for an infinitely wide Skip-$\alpha$ network is as presented in Theorem 1 where $\mathbf{E}_k$ is defined as in the theorem, but $\boldsymbol{\Sigma}_k$ is defined with $\boldsymbol{\Sigma}_0 = \mathbf{X}\mathbf{X}^T$,*

$$\boldsymbol{\Sigma}_1 = (1-\alpha)^2 \mathbf{S}\mathbf{E}_0\mathbf{S}^T + \alpha(1-\alpha)\left(\mathbf{S}\mathbf{E}_0 + \mathbf{E}_0\mathbf{S}^T\right) + \alpha^2 \mathbf{E}_0 \; and \; \boldsymbol{\Sigma}_k = (1-\alpha)^2 \mathbf{S}\mathbf{E}_{k-1}\mathbf{S}^T + \alpha^2 \tilde{\mathbf{E}}_0.$$

## 6.2 Impact of Depth in GCNs with Skip Connection

Similar to the previous section we use the NTK for Skip-PC and Skip-$\alpha$ (Corollary 3 and 4) and analyze the graph convolutions $\mathbf{S}_{sym}$ and $\mathbf{S}_{row}$ under the same considerations detailed in Section 5. Since, $\mathbf{S}_{adj}$ and $\mathbf{S}_{col}$ are theoretically worse and not popular in practice, we do not consider them for the skip connection analysis. The linear orthonormal feature NTK, $\boldsymbol{\Theta}^{(d)}$, for depth $d$ is same as $\boldsymbol{\Theta}_{lin}^{(d)}$ with changes to $\boldsymbol{\Sigma}_k$ as follows,

Skip-PC: $\boldsymbol{\Sigma}_k = \mathbf{S}^k\mathbf{S}^{kT} + \mathbf{S}\mathbf{S}^T$,

Skip-$\alpha$: $\boldsymbol{\Sigma}_k = (1-\alpha)^{2k}\mathbf{S}^k\mathbf{S}^{kT} + \alpha(1-\alpha)^{2k-1}\mathbf{S}^{k-1}\left(\mathbf{S} + \mathbf{S}^T\right)\mathbf{S}^{k-1^T} + \alpha^2 \sum_{l=1}^{k-1}(1-\alpha)^{2l}\mathbf{S}^l\mathbf{S}^{lT} + \alpha^2\mathbf{I}_n$.

We derive the population NTK $\tilde{\boldsymbol{\Theta}}^{(d)}$ and, for convenience, only state the result as $d \to \infty$ in the following theorems. Expressions for fixed $d$ are presented in Appendices B.5 and B.6.

**Theorem 5 (Class Seperability of Population NTK for Skip-PC $\zeta_{PC}^{(\infty)}$ as $d \to \infty$)** *Under the assumptions of Theorem 4,*

$$\zeta_{PC,sym}^{(\infty)} = \frac{16\tau^2 r^2}{n^2(cn)(1-r^2)}, \quad and \quad \zeta_{PC,row}^{(\infty)} = \frac{8\gamma r^2}{(cn)^2(1-r^2)} \tag{4}$$

**Theorem 6 (Class Seperability of Population NTK for Skip-$\alpha$ $\zeta_\alpha^{(\infty)}$ as $d \to \infty$)** *Under the assumptions of Theorem 4,*

$$\zeta_{\alpha,sym}^{(\infty)} = \frac{16\tau^2\alpha^2}{(cn)n^2\left(1-(1-\alpha)^2 r^2\right)}\left(\frac{1}{1-r^2}\right), \quad and \quad \zeta_{\alpha,row}^{(\infty)} = \frac{8\gamma\alpha^2}{(cn)^2\left(1-(1-\alpha)^2 r^2\right)}\left(\frac{1}{1-r^2}\right). \tag{5}$$

Theorems 5 and 6 present the class separability of population NTKs of $\mathbf{S}_{sym}$ and $\mathbf{S}_{row}$ for Skip-PC and Skip-$\alpha$, respectively. Similar to Theorem 4, assumptions on $\boldsymbol{\pi}$ in above theorems is to simplify the results. Note that $\mathbf{S}_{row}$ is better than $\mathbf{S}_{sym}$ in the case of skip connections as well due to the independence on $\boldsymbol{\pi}$ and the underlying block structures are well preserved in $\mathbf{S}_{row}$. The theorems show that the class separation in the kernel is *not zero* even at infinite depth for both Skip-PC and Skip-$\alpha$. In fact, in the case of large $n$ and $d \to \infty$, it is $\mathcal{O}\left(\frac{r^2}{n}\right)$ and $\mathcal{O}\left(\frac{\alpha^2}{n\left(1-(1-\alpha)^2 r^2\right)}\right)$ for Skip-PC and Skip-$\alpha$, respectively, since $\tau$ and $\gamma$ are $\mathcal{O}(n)$. Furthermore, to understand the role of skip connections, we plot in Figure 6 the gap between in-class and out-of-class blocks at infinite depth for different values of true class separability $r$ and small and large graph setting, for vanilla linear GCN, Skip-PC and Skip-$\alpha$ using Corollary 1, Theorems 5–6, respectively. The plot clearly shows that the block difference is away from 0 for both the skip connections in both the small and large $n$ cases given a reasonable true separation $r$, wheras the block difference in vanilla GCN is zero for small $n$ and large $n$ cases. Thus this analytical plot shows that *the class information is retained in skip connections even at infinite depth.*

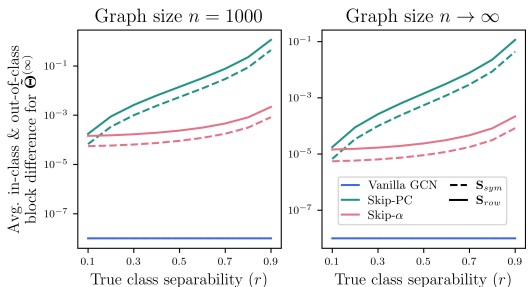
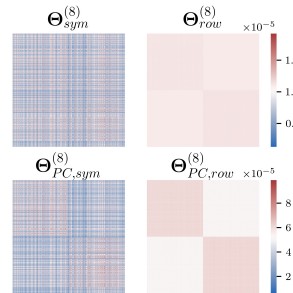

Figure 6: **Skip connection retains class information even at infinite depth. Left:** average in-class and out-of-class block difference at $d = \infty$ for small and large $n$ and different true class separability $r$ (in log scale). **Heatmaps:** exact NTKs $\boldsymbol{\Theta}^{(8)}$ for $\mathbf{S}_{sym}$ and $\mathbf{S}_{row}$ for linear GCN and Skip-PC.

### 6.3 Numerical Validation for Random Graphs

We validate our theoretical result using the same setup detailed in Section 5.2, and compute the exact NTKs for Skip-PC and Skip-$\alpha$ for both $\mathbf{S}_{sym}$ and $\mathbf{S}_{row}$. We show the result on homophilic graphs but they equally extend to the heterophilic case. While $\mathbf{S}_{sym}$ has no class information for depth=8 in vanilla GCN, it is retained reasonably in Skip-PC (right of Figure 6 column 1). In the case of $\mathbf{S}_{row}$, we clearly observe the blocks in both cases with more prevalent gap in Skip-PC illustrating our theoretical results (right of Figure 6 column 2). Similar observation is made for Skip-$\alpha$ despite considering $\mathbf{X}\mathbf{X}^T = \mathbf{I}_n$ as the model interpolates with the feature, and is discussed in Appendix C.3. Validation of the results for heterophily graphs is also included in Appendix C.3. While both $\mathbf{S}_{sym}$ and $\mathbf{S}_{row}$ retain the class information in larger depths, we observe that the degree correction plays a significant role in $\mathbf{S}_{sym}$ as elucidated in our theoretical analysis.

## 7 Empirical Analysis on Real Data

In this section, we explore how well the theoretical results translate to real dataset *Cora* with features, that is, $\mathbf{X}\mathbf{X}^T \neq \mathbf{I}_n$ and $\mathbf{A} \neq \mathbf{M}$. We consider multi-class node classification for Cora ($K = 7$). The NTKs for linear and ReLU GCNs, and GCN with Skip-PC are illustrated in Figure 7. Experimental details and additional results for Skip-$\alpha$ and *Citeseer* are in C.4 and Appendices C.5, respectively. We make the following observations from the experiments that validate the theory even in a much relaxed setting: (i) clear block structures show up in both GCN with and without skip connections for $\mathbf{S}_{row}$, thus illustrating that the class information is well retained by $\mathbf{S}_{row}$ than $\mathbf{S}_{sym}$; (ii) linear and ReLU GCNs show similar class preservation qualitatively. Thus, although the theoretical result is based on DC-SBM with mild assumptions, the conclusions hold reasonably well in real settings on real datasets as well.

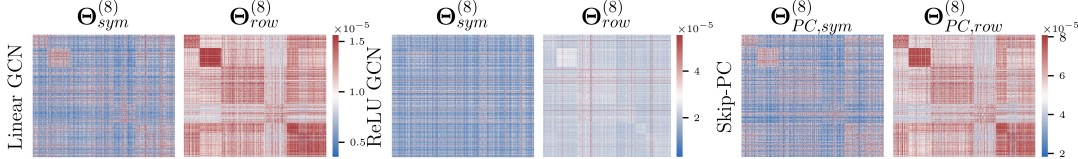

Figure 7: **Evaluation on Cora dataset.** Heatmaps show exact NTKs $\tilde{\boldsymbol{\Theta}}^{(8)}$ for linear, ReLU and Skip-PC GCNs for both symmetric and row normalized adjacency.

## 8 Discussion

**Related Work.** While GNNs are extensively used in practice, their understanding is limited, and the analysis is mostly restricted to empirical approaches (Bojchevski et al., 2018; Zhang et al., 2018; Ying et al., 2018; Wu et al., 2020). Beyond empirical methods, rigorous theoretical analysis using *learning theoretical*

*bounds* such as VC Dimension, Scarselli et al. (2018), PAC-Bayes Liao et al. (2021), Lipschitzness analysis (Tang & Liu, 2023), or sample complexity using graph topology sampling (Li et al., 2022) are propounded. Rademacher Complexity bounds (Garg et al., 2020; Esser et al., 2021) show that normalized graph convolution is beneficial, but those works do not provide insight on the influence of different normalizations on the GCN performance. Another possible tool is the NTK using which interesting theoretical insights in deep neural networks are derived (e.g. (Du et al., 2019a)). In the context of GNNs, Du et al. (2019b) derives the NTK in the supervised setting (each graph is a data instance to be classified) and empirically studies the NTK performance, however does not extend it to a theoretical analysis, and Krishnagopal & Ruiz (2023) uses Graph NTK to study convergence of large graphs. In contrast, we derive the NTK in the *semi-supervised* setting for GCN with and without skip connections, and use it to further theoretically analyze the influence of different convolutions with respect to over-smoothing. Theoretical studies (Oono & Suzuki, 2019; Cai & Wang, 2020) show that over-smoothing causes the expressive power of GNNs to decrease exponentially with depth, while Keriven (2022) proves that in linear GNNs a finite number of convolutions improves learning before over-smoothing kicks in. On the other hand, Cong et al. (2021) argues that over-smoothing does not necessarily happen in practice, and a deeper model is provably expressive. While over-smoothing and role of skip connections in GNNs are theoretically analyzed in some works (Esser et al., 2021), the influence of different convolutions that causes over-smoothing and their interplay with skip connections is not studied. For a comprehensive theory survey see Jegelka (2022).

**Conclusion.** The performance of GCNs is significantly influenced by the architecture choices, but existing learning theoretic bounds for GCNs do not provide insights specifically into the representation power of the graph convolutions and the influence of activation functions. We present a NTK based analysis that characterizes different convolutions, thereby proving the strong representation power of $\mathbf{S}_{row}$ in community detection and explaining why $\mathbf{S}_{row}$, and to some extent $\mathbf{S}_{sym}$, are preferred in practice (Theorem 4). In contrast to applying spectral analysis of the convolutions to explain over-smoothing, our explicit characterization of the network provides more exact quantification of the impact of over-smoothing in deep GCNs (Corollary 1, see Figures 3 and 4). In addition, the NTKs for GCNs with skip connections enable precise understanding of the role of skip connections in countering the over-smoothing effect (Theorems 5–6). Another value addition of our analysis is the exact quantification of the role of non-linearity (Theorem 3). While the DC-SBM assumption may seem restrictive, it is important to note that the impact of depth is derived for different convolutions exactly, therefore, making our result stronger and more precise than a general comment on the effect of over-smoothing resulting from these convolutions. Moreover, the experiments on *Cora* and *Citeseer* show that the general trends of our theoretical results extend beyond DC-SBM, although formally characterizing such behavior is difficult without model assumptions.

**Possible extensions.** *(i) Theoretical Analysis.* Considering random $\mathbf{A}$ would be more precise, but the concentration inequalities for NTK is more complex than those for Laplacians. We note that our analysis could be extended by considering feature information ($\mathbf{X}\mathbf{X}^T \neq \mathbf{I}_n$) using Contextual Stochastic Block Model as discussed in Appendix B.9, which would require more involved analysis but could provide further insights into GCNs, such as interplay between graph and feature information. *(ii) Graph Models.* The present NTK based setup allows for the analysis of different graphs having homophilic, heterophilic and core-periphery structures, and can be extended to other graph generating processes. *(iii) GCN Models.* Furthermore, the general formulation of NTK for vanilla GCNs (Theorem 1) and with skip connections (Corollaries 3–4) can be used for analyzing any new convolutions like topological structure preserving convolutions, for obtaining a rigorous understanding of GCNs by deriving statistical consistency results or information theoretic limits, as well as for theoretical analysis of other graph learning problems, such as link prediction. *(iv) Analysis.* We consider class separability as the main measure to compare different NTKs. However while we empirically observe that this measure captures the overall main trends in the MSE and accuracy, there are also cases where the measure does not capture all the trends. Therefore, we leave analyzing further ways to characterize the connection between changes in the NTK and the performance of the neural network for future study.

## 9 Acknowledgment

This work has been supported by projects from the German Research Foundation (Research Training Group GRK 2428 and Priority Program SPP 2298, project GH 257/2-1).

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

## A  Other Related Works

In contrast to the infinite width analysis, mean field limit analysis of finitely wide neural networks is conducted for various architectures at initialization (Poole et al., 2016; Schoenholz et al., 2017; Yang & Schoenholz, 2017; Xiao et al., 2018; Chen et al., 2018a; Gilboa et al., 2019; Xiao et al., 2020). This analysis resorts to initializing the weights such that the variance of weights in every layer is scaled down by the number of neurons in the layer so that the input contribution of each neuron in the layer from the activations of the previous layer remains $\mathcal{O}(1)$. The primary objective of these works is to study the trainability, generalization and expressivity aspects of the neural networks. Poole et al. (2016) shows that the networks with larger depths have the capacity to express highly non linear functions, rather than larger widths. This is extended to deriving conditions for the trainability of extremely deep neural networks in Schoenholz et al. (2017). Using similar analysis, Yang & Schoenholz (2017) shows exponential input space collapse and vanishing/exploding gradients for deep feedforward networks, whereas it becomes subexponential, even polynomial in some cases for residual connections, and Hayou et al. (2019) derives initialization parameters for different activations to accelerate training. Consequently, better initialization schemes for trainability for extremely deep neural networks based on the conditioning of input-output Jacobian matrix are established for Convolutional Neural Networks (Xiao et al., 2018), Recurrent Neural Networks and Long Short Term Memory Networks Chen et al. (2018a); Gilboa et al. (2019). Interestingly, Xiao et al. (2020) studies the trainability and generalization of networks using the condition number of the NTK and the NTK predictor, and shows that the trainability and generalizability are at odds in very wide and deep networks. In the context of GNNs, Kawamoto et al. (2018) extends the mean field analysis to graph partitioning, however exploring the potential of the analysis is still nascent.

## B  Mathematical derivations and proofs

We first derive the NTK (Theorem 1) for GCN defined in (2) and prove Theorems 2, 4, 5 and 6, Corollaries 1, 2, 3 and 4 by considering linear GCN and computing the population NTK $\tilde{\boldsymbol{\Theta}}^{(d)}$ for different graph convolutions $\mathbf{S}$. We then derive Theorem 3 for ReLU GCN similar to the analysis of linear GCN. We represent the $u$-th row of a matrix $\mathbf{M}$ as $\mathbf{M}_{u\cdot}$, and use $\mathbf{1}_n$ to denote a vector of $n$ dimension with all 1s and $\hat{\mathbf{1}}_n$ for a vector of $n$ dimension with $-1$ as first $\frac{n}{2}$ entries and $+1$ as the remaining $\frac{n}{2}$ entries, and $\mathbf{1}_{n\times n}$ for the $n \times n$ matrix of ones.

### B.1  Theorem 1: NTK for Vanilla GCN

We rewrite the GCN $F_{\mathbf{W}}(\mathbf{X}, \mathbf{S})$ defined in (2) using the following recursive definitions:

$$\mathbf{G}_1 = \mathbf{SX}, \qquad \mathbf{G}_i = \sqrt{\frac{c_\sigma}{h_{i-1}}}\mathbf{S}\sigma(\mathbf{F}_{i-1}) \;\forall i \in \{2,\ldots,d+1\}, \quad \mathbf{F}_i = \mathbf{G}_i\mathbf{W}_i \;\forall i \in [d+1]. \tag{6}$$

Thus, $F_{\mathbf{W}}(\mathbf{X}, \mathbf{S}) = \mathbf{F}_{d+1}$. Since all the output neurons behave similarly in the infinite width limit, we consider $W_{d+1}$ to be $h \times 1$ and using the definitions in (6), the gradient with respect to $\mathbf{W}_i$ of node $u$ is

$$\left(\frac{\partial F_{\mathbf{W}}(\mathbf{X}, \mathbf{S})}{\partial \mathbf{W}_i}\right)_u = (\mathbf{G}_i)^T(\mathbf{B}_i)_u \quad \text{with} \quad (\mathbf{B}_i)_u = \begin{cases} (\mathbf{1}_n)_u & \text{if } i = d+1 \\ \sqrt{\frac{c_\sigma}{h_i}}(\mathbf{S})_u^T(\mathbf{B}_{d+1})_u\mathbf{W}_{d+1}^T \odot (\dot\sigma(\mathbf{F}_i))_{u\cdot} & \text{if } i = d \\ \sqrt{\frac{c_\sigma}{h_i}}\mathbf{S}^T(\mathbf{B}_{i+1})_u\mathbf{W}_{i+1}^T \odot (\dot\sigma(\mathbf{F}_i))_{u\cdot} & \text{if } i < d \end{cases} \tag{7}$$

where $(\mathbf{B}_i)_u \in \mathbb{R}^{n \times h_i}$. We derive the NTK, as defined in (1), using the recursive definition of $F_{\mathbf{W}}(\mathbf{X}, \mathbf{S})$ in (6) and its derivative in (7). Note that the derivatives in (7) are computed for every node output following the approach in Arora et al. (2019), hence $\left(\frac{\partial F_{\mathbf{W}}(\mathbf{X},\mathbf{S})}{\partial \mathbf{W}_i}\right)_u \in \mathbb{R}^{h_{i-1} \times h_i}$. We give the gradients in B.2.

**Co-variance between Nodes.** We will first derive the co-variance matrix of size $n \times n$ for each layer comprising of co-variance between any two nodes $u$ and $v$. The co-variance between $u$ and $v$ in $\mathbf{F}_1$ and $\mathbf{F}_i$ are derived below. We denote $u$-th row of matrix $\mathbf{Z}$ as $\mathbf{Z}_{u\cdot}$ throughout our proofs.

$$\mathbb{E}\left[\left(\mathbf{F}_1\right)_{uk}\left(\mathbf{F}_1\right)_{vk'}\right] = \mathbb{E}\left[\left(\mathbf{G}_1\mathbf{W}_1\right)_{uk}\left(\mathbf{G}_1\mathbf{W}_1\right)_{vk'}\right]$$

$$= \mathbb{E}\left[\sum_{r=1}^{h_0}\left(\mathbf{G}_1\right)_{ur}\left(\mathbf{W}_1\right)_{rk}\sum_{s=1}^{h_0}\left(\mathbf{G}_1\right)_{vs}\left(\mathbf{W}_1\right)_{sk'}\right] \stackrel{(\mathbf{W}_1)_{xy}\sim\mathcal{N}(0,1)}{=} 0 \quad ; \text{ if } r \neq s \text{ or } k \neq k'$$

$$\mathbb{E}\left[\left(\mathbf{F}_1\right)_{uk}\left(\mathbf{F}_1\right)_{vk}\right] \stackrel{r\equiv s}{\underset{k=k'}{=}} \mathbb{E}\left[\sum_{r=1}^{h_0}\left(\mathbf{G}_1\right)_{ur}\left(\mathbf{G}_1\right)_{vr}\left(\mathbf{W}_1\right)_{rk}^2\right]$$

$$\stackrel{(\mathbf{W}_1)_{xy}\sim\mathcal{N}(0,1)}{=} \sum_{r=1}^{h_0}\left(\mathbf{G}_1\right)_{ur}\left(\mathbf{G}_1\right)_{vr} = \langle\left(\mathbf{G}_1\right)_{u.},\left(\mathbf{G}_1\right)_{v.}\rangle \tag{8}$$

$$\mathbb{E}\left[\left(\mathbf{F}_i\right)_{uk}\left(\mathbf{F}_i\right)_{vk}\right] \stackrel{r\equiv s}{\underset{k=k'}{=}} \mathbb{E}\left[\sum_{r=1}^{h_{i-1}}\left(\mathbf{G}_i\right)_{ur}\left(\mathbf{G}_i\right)_{vr}\left(\mathbf{W}_i\right)_{rk}^2\right]$$

$$\stackrel{(\mathbf{W}_i)_{xy}\sim\mathcal{N}(0,1)}{=} \sum_{r=1}^{h_{i-1}}\left(\mathbf{G}_i\right)_{ur}\left(\mathbf{G}_i\right)_{vr} = \langle\left(\mathbf{G}_i\right)_{u.},\left(\mathbf{G}_i\right)_{v.}\rangle \tag{9}$$

Evaluating (8) and (9) in terms of the graph in the following,

$$(8): \quad \langle\left(\mathbf{G}_1\right)_{u.},\left(\mathbf{G}_1\right)_{v.}\rangle = \langle\left(\mathbf{SX}\right)_{u.},\left(\mathbf{SX}\right)_{v.}\rangle = \mathbf{S}_{u.}\mathbf{XX}^T\mathbf{S}_{.v}^T = \left(\mathbf{\Sigma}_1\right)_{uv} \tag{10}$$

$$(9): \quad \langle\left(\mathbf{G}_i\right)_{u.},\left(\mathbf{G}_i\right)_{v.}\rangle = \frac{c_\sigma}{h_{i-1}}\langle\left(\mathbf{S}\sigma(\mathbf{F}_{i-1})\right)_{u.},\left(\mathbf{S}\sigma(\mathbf{F}_{i-1})\right)_{v.}\rangle$$

$$= \frac{c_\sigma}{h_{i-1}}\sum_{k=1}^{h_{i-1}}\left(\mathbf{S}\sigma(\mathbf{F}_{i-1})\right)_{uk}\left(\mathbf{S}\sigma(\mathbf{F}_{i-1})\right)_{vk}$$

$$\stackrel{h_{i-1}\to\infty}{=} c_\sigma\mathbb{E}\left[\left(\mathbf{S}\sigma(\mathbf{F}_{i-1})\right)_{uk}\left(\mathbf{S}\sigma(\mathbf{F}_{i-1})\right)_{vk}\right] \qquad ; \text{ law of large numbers}$$

$$= c_\sigma\mathbb{E}\left[\left(\sum_{r=1}^{n}\mathbf{S}_{ur}\sigma\left(\mathbf{F}_{i-1}\right)_{rk}\right)\left(\sum_{s=1}^{n}\mathbf{S}_{vs}\sigma\left(\mathbf{F}_{i-1}\right)_{sk}\right)\right]$$

$$= c_\sigma\mathbb{E}\left[\sum_{r=1}^{n}\sum_{s=1}^{n}\mathbf{S}_{ur}\mathbf{S}_{vs}\sigma\left(\mathbf{F}_{i-1}\right)_{rk}\sigma\left(\mathbf{F}_{i-1}\right)_{sk}\right]$$

$$\stackrel{(a)}{=} \sum_{r=1}^{n}\sum_{s=1}^{n}\mathbf{S}_{ur}\left(\mathbf{E}_{i-1}\right)_{rs}\mathbf{S}_{sv}^T = \mathbf{S}_{u.}\mathbf{E}_{i-1}\mathbf{S}_{.v}^T = \left(\mathbf{\Sigma}_i\right)_{uv} \tag{11}$$

$(a)$: using $\mathbb{E}\left[\left(\mathbf{F}_{i-1}\right)_{rk}\left(\mathbf{F}_{i-1}\right)_{sk}\right] = \left(\mathbf{\Sigma}_{i-1}\right)_{rs}$ and the definition of $\mathbf{E}_{i-1}$ in Theorem 1.

**NTK for Vanilla GCN.** Let us first evaluate the tangent kernel component from $\mathbf{W}_k$ respective to nodes $u$ and $v$. The following two results are needed to derive it. To compute the NTK we need to evaluate the sum of all parameters gradient dot product between two nodes $u$ and $v$. To do so, we first evaluate $\left\langle\left(\frac{\partial\mathbf{F}}{\partial\mathbf{W}_k}\right)_u,\left(\frac{\partial\mathbf{F}}{\partial\mathbf{W}_k}\right)_v\right\rangle$ in the following.

$$\left\langle\left(\frac{\partial\mathbf{F}}{\partial\mathbf{W}_k}\right)_u,\left(\frac{\partial\mathbf{F}}{\partial\mathbf{W}_k}\right)_v\right\rangle = \sum_{i=1,j=1}^{h_{k-1},h_k}\left(\left(\frac{\partial\mathbf{F}}{\partial\mathbf{W}_k}\right)_u\right)_{ij}\left(\left(\frac{\partial\mathbf{F}}{\partial\mathbf{W}_k}\right)_v\right)_{ij}$$

$$= \sum_{i=1,j=1}^{h_{k-1},h_k}\left(\mathbf{G}_k^T\left(\mathbf{B}_k\right)_u\right)_{ij}\left(\mathbf{G}_k^T\left(\mathbf{B}_k\right)_v\right)_{ij}$$

$$= \sum_{i=1,j=1}^{h_{k-1},h_k}\sum_{a=1,b=1}^{n,n}\left(\mathbf{G}_k^T\right)_{ia}\left(\left(\mathbf{B}_k\right)_u\right)_{aj}\left(\mathbf{G}_k^T\right)_{ib}\left(\left(\mathbf{B}_k\right)_v\right)_{bj}$$

$$= \sum_{j=1}^{h_k} \sum_{a=1,b=1}^{n,n} \frac{c_\sigma}{h_k} \left(\boldsymbol{S}^T (\mathbf{B}_{k+1})_u \mathbf{W}_{k+1}^T\right)_{aj} \left(\dot\sigma (\mathbf{F}_k)\right)_{aj} \left(\mathbf{G}_k \mathbf{G}_k^T\right)_{ab} \left(\boldsymbol{S}^T (\mathbf{B}_{k+1})_u \mathbf{W}_{k+1}^T\right)_{bj} \left(\dot\sigma (\mathbf{F}_k)\right)_{bj} \quad (12)$$

$$= \sum_{j=1,l=1,m=1}^{h_k,h_{k+1},h_{k+1}} \sum_{a=1,b=1}^{n,n} \frac{c_\sigma}{h_k} \left(\boldsymbol{S}^T (\mathbf{B}_{k+1})_u\right)_{al} \left(\mathbf{W}_{k+1}^T\right)_{lj} \left(\dot\sigma (\mathbf{F}_k)\right)_{aj} \left(\mathbf{G}_k \mathbf{G}_k^T\right)_{ab} \left(\boldsymbol{S}^T (\mathbf{B}_{k+1})_u\right)_{bm} \left(\mathbf{W}_{k+1}^T\right)_{mj} \left(\dot\sigma (\mathbf{F}_k)\right)_{bj}$$

$$\overset{\substack{h_k \to \infty \\ h_{k+1} \to \infty}}{=} c_\sigma \sum_{j=1,l=1}^{h_k,h_{k+1}} \sum_{a=1,b=1}^{n,n} \left(\boldsymbol{S}^T (\mathbf{B}_{k+1})_u\right)_{al} \left(\dot\sigma (\mathbf{F}_k)\right)_{aj} \left(\mathbf{G}_k \mathbf{G}_k^T\right)_{ab} \left(\boldsymbol{S}^T (\mathbf{B}_{k+1})_u\right)_{bl} \left(\dot\sigma (\mathbf{F}_k)\right)_{bj}$$

$$= c_\sigma \sum_{l=1}^{h_{k+1}} \sum_{a=1,b=1}^{n,n} \left(\boldsymbol{S}^T (\mathbf{B}_{k+1})_u\right)_{al} \left(\boldsymbol{S}^T (\mathbf{B}_{k+1})_u\right)_{bl} \left(\mathbf{G}_k \mathbf{G}_k^T\right)_{ab} \mathbb{E}\left[\left(\dot\sigma (\mathbf{F}_k) \dot\sigma (\mathbf{F}_k)^T\right)_{ab}\right]$$

$$\overset{(b)}{=} \sum_{l=1}^{h_{k+1}} \left(\left(\boldsymbol{S}^T (\mathbf{B}_{k+1})_u\right)^T \left(\mathbf{G}_k \mathbf{G}_k^T \odot \dot{\mathbf{E}}_k\right) \left(\boldsymbol{S}^T (\mathbf{B}_{k+1})_u\right)\right)_{ll}$$

$$= \mathrm{tr}((\mathbf{B}_{k+1})_u^T \mathbf{S} \left(\boldsymbol{\Sigma}_k \odot \dot{\mathbf{E}}_k\right) \mathbf{S}^T (\mathbf{B}_{k+1})_v)$$

$$\overset{(c)}{=} \mathrm{tr}((\mathbf{B}_{d+1})_u^T \mathbf{S}_u \left(\ldots \mathbf{S} \left(\mathbf{S} \left(\boldsymbol{\Sigma}_k \odot \dot{\mathbf{E}}_k\right) \mathbf{S}^T \odot \dot{\mathbf{E}}_{k+1}\right) \mathbf{S}^T \odot \ldots \odot \dot{\mathbf{E}}_d\right) \mathbf{S}_v^T (\mathbf{B}_{d+1})_v)$$

$$= \mathbf{S}_{u.} \left(\ldots \mathbf{S} \left(\mathbf{S} \left(\boldsymbol{\Sigma}_k \odot \dot{\mathbf{E}}_k\right) \mathbf{S}^T \odot \dot{\mathbf{E}}_{k+1}\right) \mathbf{S}^T \odot \ldots \odot \dot{\mathbf{E}}_d\right) \mathbf{S}_{v.}^T \quad (13)$$

(b): $c_\sigma \mathbb{E}\left[\left(\dot\sigma (\mathbf{F}_k) \dot\sigma (\mathbf{F}_k)^T\right)_{ab}\right] = \left(\dot{\mathbf{E}}_k\right)_{ab}$.

(c): Expanding $\mathbf{B}_{k+1}$ will result in the expression similar to (12), and repeated expansion until $\mathbf{B}_{d+1}$. The final equation is obtained by substituting $(\mathbf{B}_{d+1})_u = 1$ from its definition in (3).

Extending (13) to all $n$ nodes which will result in $n \times n$ matrix, we get

$$\left\langle \frac{\partial \mathbf{F}}{\partial \mathbf{W}_k}, \frac{\partial \mathbf{F}}{\partial \mathbf{W}_k} \right\rangle = \mathbf{S} \left(\ldots \mathbf{S} \left(\mathbf{S} \left(\boldsymbol{\Sigma}_k \odot \dot{\mathbf{E}}_k\right) \mathbf{S}^T \odot \dot{\mathbf{E}}_{k+1}\right) \mathbf{S}^T \odot \ldots \odot \dot{\mathbf{E}}_d\right) \mathbf{S}^T$$

$$\underset{\mathbf{W}_k}{\mathbb{E}}\left[\left\langle \frac{\partial \mathbf{F}}{\partial \mathbf{W}_k}, \frac{\partial \mathbf{F}}{\partial \mathbf{W}_k} \right\rangle\right] = \mathbf{S} \left(\ldots \mathbf{S} \left(\mathbf{S} \left(\boldsymbol{\Sigma}_k \odot \dot{\mathbf{E}}_k\right) \mathbf{S}^T \odot \dot{\mathbf{E}}_{k+1}\right) \mathbf{S}^T \odot \ldots \odot \dot{\mathbf{E}}_d\right) \mathbf{S}^T \quad (14)$$

Finally, NTK $\boldsymbol{\Theta}$ is,

$$\boldsymbol{\Theta} = \sum_{k=1}^{d+1} \underset{\mathbf{W}_k}{\mathbb{E}}\left[\left\langle \frac{\partial \mathbf{F}}{\partial \mathbf{W}_k}, \frac{\partial \mathbf{F}}{\partial \mathbf{W}_k} \right\rangle\right]$$

$$= \sum_{k=1}^{d+1} \mathbf{S} \left(\ldots \mathbf{S} \left(\mathbf{S} \left(\boldsymbol{\Sigma}_k \odot \dot{\mathbf{E}}_k\right) \mathbf{S}^T \odot \dot{\mathbf{E}}_{k+1}\right) \mathbf{S}^T \odot \ldots \odot \dot{\mathbf{E}}_d\right) \mathbf{S}^T \quad (15)$$

with definition of $\boldsymbol{\Sigma}_k$ and $\dot{\mathbf{E}}_k$ mentioned in the theorem. $\qquad\square$

## B.2 Gradients of functions with scalar output

We list here the aggregation of gradients for different functions that enable deriving the equation (7). The following $\frac{\partial f}{\partial \mathbf{W}}$ are derived assuming $f \in \mathbb{R}$. Hence the derivative will be of same dimension as $\mathbf{W}$.

$$\frac{\partial \mathbf{XW}}{\partial \mathbf{W}} = \mathbf{X}^T \mathbf{1} \quad ; \qquad \frac{\partial \sigma(\mathbf{XW})}{\partial \mathbf{W}} = \mathbf{X}^T \dot{\sigma}(\mathbf{XW})$$

$$\frac{\partial \mathbf{XWY}}{\partial \mathbf{W}} = \mathbf{X}^T \mathbf{1} \mathbf{Y}^T \quad ; \qquad \frac{\partial \sigma(\mathbf{XWY})}{\partial \mathbf{W}} = \mathbf{X}^T \dot{\sigma}(\mathbf{XWY}) \mathbf{Y}^T$$

$$\frac{\partial \mathbf{Z}\sigma(\mathbf{XW})\mathbf{Y}}{\partial \mathbf{W}} = \mathbf{X}^T \left( \mathbf{Z}^T \mathbf{1} \mathbf{Y}^T \odot \dot{\sigma}(\mathbf{XW}) \right)$$

$$\frac{\partial \sigma(\mathbf{Z}_1 \sigma(\mathbf{Z}_2 \sigma(\mathbf{XW})\mathbf{Y}_1)\mathbf{Y}_2)}{\partial \mathbf{W}} = \mathbf{X}^T \left( \mathbf{Z}_2^T \left( \mathbf{Z}_1^T \dot{\sigma}(\mathbf{Z}_1 \sigma(\mathbf{Z}_2 \sigma(\mathbf{XW})\mathbf{Y}_1)\mathbf{Y}_2)\mathbf{Y}_2^T \odot \dot{\sigma}(\mathbf{Z}_2 \sigma(\mathbf{XW})\mathbf{Y}_1) \right) \mathbf{Y}_1^T \odot \dot{\sigma}(\mathbf{XW}) \right)$$

In the above, all $\mathbf{1}$ are scalars. These derivatives are used to derive (7).

### B.3 Theorems 2, 4 and Corollary 1: Population NTK $\tilde{\Theta}$ for Different Convolutions S

We consider linear GCN with Assumption 1, that is, orthonormal features and Assumption 2. We derive it generally without the assumption on $\gamma$. We first prove it for $K = 2$ and then extend it to $K$ classes. We consider that all nodes are sorted per class for ease of analysis which implies $\mathbf{A}$ is a $n \times n$ matrix with $p\pi_i\pi_j$ entries in $[1, \frac{n}{2}][1, \frac{n}{2}]$ and $[\frac{n}{2}+1, n][\frac{n}{2}+1, n]$ blocks and $q\pi_i\pi_j$ entries in $[1, \frac{n}{2}][\frac{n}{2}+1, n]$ and $[\frac{n}{2}+1, n][1, \frac{n}{2}]$ blocks. Therefore,

$$\mathbf{A} = \boldsymbol{\pi}\boldsymbol{\pi}^T \odot \left( \frac{p+q}{2}\mathbf{1}\mathbf{1}^T + \frac{p-q}{2}\hat{\mathbf{1}}\hat{\mathbf{1}}^T \right)$$

$$= \frac{p+q}{2}\boldsymbol{\pi}\boldsymbol{\pi}^T + \frac{p-q}{2}\hat{\boldsymbol{\pi}}\hat{\boldsymbol{\pi}}^T \tag{16}$$

where the entries of $\hat{\boldsymbol{\pi}}$ are $-\pi_i \,\forall\, i \in [1, \frac{n}{2}]$ and $+\pi_i \,\forall\, i \in [\frac{n}{2}+1, n]$. The degree matrix $\mathbf{D}$ is $\mathbf{D} = \frac{(p+q)cn}{2}\text{diag}(\boldsymbol{\pi})$.

#### B.3.1 Symmetric Degree Normalized Adjacency $\mathbf{S}_{sym}$

Now, lets compute $\mathbf{S}_{sym}$ using $\mathbf{A}$ (16) and its degree matrix $\mathbf{D}$.

$$\mathbf{S}_{sym} = \mathbf{D}^{-\frac{1}{2}}\mathbf{A}\mathbf{D}^{-\frac{1}{2}}$$

$$= \frac{2}{(p+q)\,cn}\text{diag}(\boldsymbol{\pi})^{-\frac{1}{2}}\left( \frac{p+q}{2}\boldsymbol{\pi}\boldsymbol{\pi}^T + \frac{p-q}{2}\hat{\boldsymbol{\pi}}\hat{\boldsymbol{\pi}}^T \right)\text{diag}(\boldsymbol{\pi})^{-\frac{1}{2}}$$

$$= \frac{1}{cn}\left( \boldsymbol{\pi}^{\frac{1}{2}}\boldsymbol{\pi}^{\frac{1}{2}T} + \frac{p-q}{p+q}\hat{\boldsymbol{\pi}}^{\frac{1}{2}}\hat{\boldsymbol{\pi}}^{\frac{1}{2}T} \right)$$

$$= \begin{bmatrix} \frac{\sqrt{\pi_1}}{\sqrt{cn}} & -\frac{\sqrt{\pi_1}}{\sqrt{cn}} \\ \vdots & \vdots \\ \frac{\sqrt{\pi_n}}{\sqrt{cn}} & +\frac{\sqrt{\pi_n}}{\sqrt{cn}} \end{bmatrix}_{n\times 2} \begin{bmatrix} 1 & 0 \\ 0 & r \end{bmatrix}_{2\times 2} \begin{bmatrix} \frac{\sqrt{\pi_1}}{\sqrt{cn}} & -\frac{\sqrt{\pi_1}}{\sqrt{cn}} \\ \vdots & \vdots \\ \frac{\sqrt{\pi_n}}{\sqrt{cn}} & +\frac{\sqrt{\pi_n}}{\sqrt{cn}} \end{bmatrix}_{2\times n}^T$$

$$= \mathbf{U}\boldsymbol{\Lambda}\mathbf{U}^T \tag{17}$$

Note that $\boldsymbol{\pi}^{\frac{1}{2}T}\boldsymbol{\pi}^{\frac{1}{2}} = \hat{\boldsymbol{\pi}}^{\frac{1}{2}T}\hat{\boldsymbol{\pi}}^{\frac{1}{2}} = cn$, $\boldsymbol{\pi}^{\frac{1}{2}T}\hat{\boldsymbol{\pi}}^{\frac{1}{2}} = 0$ since $\sum_{i \in \mathcal{C}_k} \pi = \frac{cn}{K}$ and $\mathbf{U}^T\mathbf{U} = \mathbf{I}_2$, thus (17) is the singular value decomposition of $\mathbf{S}_{sym}$.

To compute the population NTK $\tilde{\mathbf{\Theta}}^{(d)}_{sym}$ for linear GCN with orthonormal features, we need $\mathbf{S}^k_{sym}\mathbf{S}^{kT}_{sym}$. Using (17),

$$\mathbf{S}^k_{sym}\mathbf{S}^{kT}_{sym} \overset{(17)}{=} \mathbf{U}\mathbf{\Lambda}^{2k}\mathbf{U}^T$$

$$= \begin{bmatrix} \frac{\sqrt{\pi_1}}{\sqrt{cn}} & -\frac{\sqrt{\pi_1}}{\sqrt{cn}} \\ \vdots & \vdots \\ \frac{\sqrt{\pi_n}}{\sqrt{cn}} & +\frac{\sqrt{\pi_n}}{\sqrt{cn}} \end{bmatrix}_{n \times 2} \begin{bmatrix} 1 & 0 \\ 0 & r^{2k} \end{bmatrix}_{2\times 2} \begin{bmatrix} \frac{\sqrt{\pi_1}}{\sqrt{cn}} & -\frac{\sqrt{\pi_1}}{\sqrt{cn}} \\ \vdots & \vdots \\ \frac{\sqrt{\pi_n}}{\sqrt{cn}} & +\frac{\sqrt{\pi_n}}{\sqrt{cn}} \end{bmatrix}^T_{2 \times n}$$

$$\left(\mathbf{S}^k_{sym}\mathbf{S}^{kT}_{sym}\right)_{ij} = \left(1 + \delta_{ij}r^{2k}\right)\frac{\sqrt{\pi_i\pi_j}}{cn} \qquad\qquad\qquad ; \delta_{ij} = (-1)^{\mathbb{1}[\mathcal{C}_i \neq \mathcal{C}_j]}$$

$$\mathbf{S}^k_{sym}\mathbf{S}^{kT}_{sym} \underset{\text{notation}}{\overset{\text{matrix}}{=}} (cn)^{-1}\left[\begin{array}{c|c} \left(1+r^{2k}\right)\sqrt{\pi_i\pi_j} & \left(1-r^{2k}\right)\sqrt{\pi_i\pi_j} \\ \hline \underbrace{\left(1-r^{2k}\right)\sqrt{\pi_i\pi_j}}_{\frac{n}{2}\text{ entries}} & \underbrace{\left(1+r^{2k}\right)\sqrt{\pi_i\pi_j}}_{\frac{n}{2}\text{ entries}} \end{array}\right]_{n \times n} \tag{18}$$

Consequently, population NTK $\tilde{\mathbf{\Theta}}^{(d)}_{sym}$ for nodes $i$ and $j$ using (18) is as follows,

$$\left(\tilde{\mathbf{\Theta}}^{(d)}_{sym}\right)_{ij} = \sum_{k=1}^{d+1}\mathbf{S}^{d+1}_{sym}\mathbf{S}^{(d+1)T}_{sym}$$

$$= (d+1)\left(1+\delta_{ij}r^{2d+2}\right)\frac{\sqrt{\pi_i\pi_j}}{cn} \tag{19}$$

Hence, the average block difference of the population NTK which we refer to class separability of the kernel $\zeta^{(d)}_{sym}$ is derived with $\sum_{i=1}^n \sqrt{\pi_i}\mathbb{1}[\mathcal{C}_i = k] = \tau_k \,\forall\, k$

$$\zeta^{(d)}_{sym} = \frac{4(d+1)}{n^2(cn)}\left(\sum_{i=1}^{n/2}\sum_{j=1}^{n/2}\left(1+r^{2d+2}\right)\sqrt{\pi_i\pi_j} + \sum_{i=n/2+1}^{n}\sum_{j=n/2+1}^{n}\left(1+r^{2d+2}\right)\sqrt{\pi_i\pi_j}\right.$$

$$\left. -\sum_{i=1}^{n/2}\sum_{j=n/2+1}^{n}\left(1-r^{2d+2}\right)\sqrt{\pi_i\pi_j} - \sum_{i=n/2+1}^{n}\sum_{j=1}^{n/2}\left(1-r^{2d+2}\right)\sqrt{\pi_i\pi_j}\right)$$

$$= \frac{4(d+1)}{n^2(cn)}\left(1+r^{2d+2}\right)\left(\tau_1^2 + \tau_2^2\right) - 2\left(1-r^{2d+2}\right)\left(\tau_1\tau_2\right)$$

$$= \frac{d+1}{cn}\left(\frac{4}{n^2}\left(\tau_1 - \tau_2\right)^2 + \frac{4}{n^2}r^{2d+2}\left(\tau_1 + \tau_2\right)^2\right) \tag{20}$$

In (20), $\tau_1$ is of same order as $\tau_2$ and $\tau_1 \approx \tau_2$ for large $n$ with $\sum_{i\in\mathcal{C}_k}\pi = \frac{cn}{K}$. Hence, considering $\tau_1 = \tau_2 = \tau$, we get the block difference as $\frac{16\tau^2(d+1)}{n^2(cn)}r^{2d+2}$. It is of $\mathcal{O}(\frac{dr^{2d}}{n})$, since $\left(\tau_1 + \tau_2\right)^2$ has $n^2$ terms, each of $\mathcal{O}(1)$.

Therefore, the block difference of the population NTK $\tilde{\mathbf{\Theta}}^{(d)}_{sym}$ at $d \to \infty$ is

$$\lim_{d\to\infty}\frac{16\tau^2(d+1)}{n^2(cn)}r^{2d+2} = \lim_{d\to\infty}\frac{16\tau^2}{n^2(cn)}\frac{d+1}{r^{-(2d+2)}}$$

$$= \lim_{d\to\infty}\frac{16\tau^2}{n^2(cn)}\frac{1}{r^{-(2d+2)}\log(r)(-2)} = 0 \tag{21}$$

Apart from the block difference, we can also see that the population kernel at $ij$ is proportional to $\frac{\sqrt{\pi_i\pi_j}}{cn}$ as $d \to \infty$, thus converging to a constant kernel. Equations (19) and (21) prove the population NTK $\tilde{\mathbf{\Theta}}^{(d)}_{sym}$ and class separability of $\tilde{\mathbf{\Theta}}^{(\infty)}_{sym}$ in Theorem 4 and Corollary 1, respectively. Substituting $d = 1$ and $\forall i, \pi_i = \frac{1}{n}$, Theorem 2 can be derived. $\qquad\square$

### B.3.2 Row Degree Normalized Adjacency $\mathbf{S}_{row}$

The assumption on $\gamma$ in Assumption 2 is only to simplify the expression of population NTK for $\mathbf{S}_{row}$. We derive it without this assumption in the following. We first derive $\mathbf{S}_{row}^k \mathbf{S}_{row}^{kT}$.

$$
\begin{aligned}
\mathbf{S}_{row} &= \mathbf{D}^{-1}\mathbf{A} \\
&= \mathbf{D}^{-\frac{1}{2}}\mathbf{D}^{-\frac{1}{2}}\mathbf{A}\mathbf{D}^{-\frac{1}{2}}\mathbf{D}^{+\frac{1}{2}} \\
&= \mathbf{D}^{-\frac{1}{2}}\mathbf{U}\boldsymbol{\Lambda}\mathbf{U}^T\mathbf{D}^{+\frac{1}{2}} \\
\mathbf{S}_{row}^k &= \mathbf{D}^{-\frac{1}{2}}\mathbf{U}\boldsymbol{\Lambda}^k\mathbf{U}^T\mathbf{D}^{+\frac{1}{2}} \\
\mathbf{S}_{row}^k\mathbf{S}_{row}^{kT} &= \mathbf{D}^{-\frac{1}{2}}\mathbf{U}\boldsymbol{\Lambda}^k\mathbf{U}^T\mathbf{D}^{+\frac{1}{2}}\mathbf{D}^{+\frac{1}{2}}\mathbf{U}\boldsymbol{\Lambda}^k\mathbf{U}^T\mathbf{D}^{-\frac{1}{2}} \\
&= \mathbf{D}^{-\frac{1}{2}}\mathbf{U}\boldsymbol{\Lambda}^k\mathbf{U}^T\mathbf{D}\mathbf{U}\boldsymbol{\Lambda}^k\mathbf{U}^T\mathbf{D}^{-\frac{1}{2}} \\
&= \left(\mathbf{D}^{-\frac{1}{2}}\mathbf{U}\boldsymbol{\Lambda}^k\mathbf{U}^T\mathbf{D}^{-\frac{1}{2}}\right)\mathbf{D}^{+\frac{1}{2}}\mathbf{D}\mathbf{D}^{+\frac{1}{2}}\left(\mathbf{D}^{-\frac{1}{2}}\mathbf{U}\boldsymbol{\Lambda}^k\mathbf{U}^T\mathbf{D}^{-\frac{1}{2}}\right) \\
&= \left(\widehat{\mathbf{U}}\boldsymbol{\Lambda}^k\widehat{\mathbf{U}}^T\right)\mathbf{D}^2\left(\widehat{\mathbf{U}}\boldsymbol{\Lambda}^k\widehat{\mathbf{U}}^T\right) \qquad ; \widehat{\mathbf{U}} = \mathbf{D}^{-\frac{1}{2}}\mathbf{U} = \frac{\sqrt{2}}{cn\sqrt{p+q}}\begin{bmatrix}\mathbf{1}_n^T \\ \widehat{\mathbf{1}}_n^T\end{bmatrix}_{n\times 2}
\end{aligned}
$$

$$
\left(\mathbf{S}_{row}^k\mathbf{S}_{row}^{kT}\right)_{ij} = (cn)^{-2}\begin{cases}\left(1+r^k\right)^2\lambda + \left(1-r^k\right)^2\mu & \text{if } i \text{ and } j \in \text{class 1} \\ \left(1+r^k\right)\left(1-r^k\right)(\lambda+\mu) & \text{if } i \text{ and } j \notin \text{same class} \\ \left(1-r^k\right)^2\lambda + \left(1+r^k\right)^2\mu & \text{if } i \text{ and } j \in \text{class 2}\end{cases} ; \lambda = \sum_{s=1}^{\frac{n}{2}}\pi_s^2; \ \mu = \sum_{s=\frac{n}{2}+1}^{n}\pi_s^2
$$

$$
\mathbf{S}_{row}^k\mathbf{S}_{row}^{kT} \overset{\text{matrix not.}}{=} (cn)^{-2}\left[\begin{array}{c|c}\left(1+r^k\right)^2\lambda+\left(1-r^k\right)^2\mu & \left(1+r^k\right)\left(1-r^k\right)(\lambda+\mu) \\ \hline \underbrace{\left(1+r^k\right)\left(1-r^k\right)(\lambda+\mu)}_{\frac{n}{2}\text{ entries}} & \underbrace{\left(1-r^k\right)^2\lambda+\left(1+r^k\right)^2\mu}_{\frac{n}{2}\text{ entries}}\end{array}\right]_{n\times n} \tag{22}
$$

Note that each block is a constant and independent of individual $\pi_i$. Using (22) and the assumption $\lambda = \mu = \gamma$ in Theorem 4, the population NTK for nodes $i$ and $j$ is,

$$
\begin{aligned}
\left(\tilde{\boldsymbol{\Theta}}_{row}^{(d)}\right)_{ij} &\overset{(22)}{=} \sum_{k=1}^{d+1}\mathbf{S}_{row}^{d+1}\mathbf{S}_{row}^{(d+1)T} \\
&= (d+1)\left(1+\delta_{ij}r^{2d+2}\right)\frac{2\gamma}{(cn)^2}
\end{aligned} \tag{23}
$$

Using (23), we derive the class separability of the kernel $\zeta_{row}^{(d)}$.

$$
\zeta_{row}^{(d)} = \frac{2\gamma(d+1)}{(cn)^2}4r^{2d+2} \tag{24}
$$

Similar to (20), $\zeta_{row}^{(d)}$ is of $\mathcal{O}(\frac{dr^{2d}}{n})$ since $\gamma$ is $\mathcal{O}(n)$, and the class separability of the population NTK $\tilde{\boldsymbol{\Theta}}_{row}^{(d)}$ at $d \to \infty$ is 0. Likewise, the population kernel at $ij$ is proportional to $\frac{2\gamma}{(cn)^2}$ as $d \to \infty$, thus converging to a constant kernel proving Theorem 4 and Corollary 1, respectively. $\qquad\square$

### B.3.3 Column Normalized Adjacency $\mathbf{S}_{col}$

In this section we derive the population NTK $\tilde{\mathbf{\Theta}}_{col}^{(d)}$.

$$\mathbf{S}_{col} = \mathbf{A}\mathbf{D}^{-1}$$
$$= \mathbf{D}^{+\frac{1}{2}}\mathbf{U}\mathbf{\Lambda}\mathbf{U}^T\mathbf{D}^{-\frac{1}{2}}$$
$$\mathbf{S}_{col}^k = \mathbf{D}^{+\frac{1}{2}}\mathbf{U}\mathbf{\Lambda}^k\mathbf{U}^T\mathbf{D}^{-\frac{1}{2}}$$
$$\mathbf{S}_{col}^k\mathbf{S}_{col}^{kT} = \mathbf{D}^{+\frac{1}{2}}\mathbf{U}\mathbf{\Lambda}^k\mathbf{U}^T\mathbf{D}^{-\frac{1}{2}}\mathbf{D}^{-\frac{1}{2}}\mathbf{U}\mathbf{\Lambda}^k\mathbf{U}^T\mathbf{D}^{+\frac{1}{2}}$$
$$= \left(\tilde{\mathbf{U}}\mathbf{\Lambda}^k\tilde{\mathbf{U}}^T\right)\mathbf{D}^{-2}\left(\tilde{\mathbf{U}}\mathbf{\Lambda}^k\tilde{\mathbf{U}}^T\right) \qquad ; \tilde{\mathbf{U}} = \mathbf{D}^{+\frac{1}{2}}\mathbf{U} = \sqrt{\frac{p+q}{2}}\begin{bmatrix}\boldsymbol{\pi}^T\\\hat{\boldsymbol{\pi}}^T\end{bmatrix}_{n\times 2}$$

$$\overset{\text{matrix not.}}{=\joinrel=} \frac{n}{(cn)^2}\left[\begin{array}{c|c}\pi_i\pi_j\left(1+r^{2k}\right) & \pi_i\pi_j\left(1-r^{2k}\right)\\\hline \underbrace{\pi_i\pi_j\left(1-r^{2k}\right)}_{\frac{n}{2}\text{ entries}} & \underbrace{\pi_i\pi_j\left(1+r^{2k}\right)}_{\frac{n}{2}\text{ entries}}\end{array}\right]_{n\times n} \tag{25}$$

Therefore, $\tilde{\mathbf{\Theta}}_{col}^{(d)}$ for all $i$ and $j$ is

$$\left(\tilde{\mathbf{\Theta}}_{col}^{(d)}\right)_{ij} \overset{(25)}{=} \sum_{k=1}^{d+1}\mathbf{S}_{col}^{d+1}\mathbf{S}_{col}^{(d+1)T}$$
$$= (d+1)\left(1+\delta_{ij}r^{2d+2}\right)\frac{n\pi_i\pi_j}{(cn)^2} \tag{26}$$

Using (26) and $\sum_{i\in\mathcal{C}_k}\pi = \frac{cn}{K}$, the class separability of the kernel $\zeta_{col}^{(d)}$ is

$$\zeta_{col}^{(d)} = \frac{4(d+1)}{n}r^{2d+2} \tag{27}$$

which is of $\mathcal{O}(\frac{dr^{2d}}{n})$ and the class separability of the population NTK $\tilde{\mathbf{\Theta}}_{row}^{(d)}$ at $d \to \infty$ is 0 similar to symmetric and row normalization cases. Likewise, the population kernel at $ij$ is proportional to $\frac{n\pi_i\pi_j}{(cn)^2}$ as $d \to \infty$, thus converging to a constant kernel. Hence, equations (26) and (27) prove the population NTK $\tilde{\mathbf{\Theta}}_{col}^{(d)}$ and $\zeta_{col}^{(d)}$ in Theorem 4 and Corollary 1, respectively. $\qquad\square$

### B.3.4 Unnormalized Adjacency $\mathbf{S}_{adj}$

We can rewrite $\mathbf{A}$ as follows,

$$\mathbf{A} = \boldsymbol{\pi}\boldsymbol{\pi}^T \odot \left[\begin{array}{c|c}p & q\\\hline \underbrace{q}_{\frac{n}{2}\text{ entries}} & \underbrace{p}_{\frac{n}{2}\text{ entries}}\end{array}\right]_{n\times n}$$
$$= \begin{bmatrix}\pi_1 & & \\ & \ddots & \\ & & \pi_n\end{bmatrix}_{n\times n}\left[\begin{array}{c|c}p & q\\\hline q & p\end{array}\right]_{n\times n}\begin{bmatrix}\pi_1 & & \\ & \ddots & \\ & & \pi_n\end{bmatrix}_{n\times n} \tag{28}$$

We consider $\gamma$ assumption for the analysis of unnormalised adjacency to simplify the computation. But the result holds without this assumption.

$$\mathbf{A}^2 \overset{(28)}{=} \begin{bmatrix} \pi_1 & & \\ & \ddots & \\ & & \pi_n \end{bmatrix} \begin{bmatrix} \left(p^2+q^2\right)\gamma & 2pq\gamma \\ \hline 2pq\gamma & \left(p^2+q^2\right)\gamma \end{bmatrix} \begin{bmatrix} \pi_1 & & \\ & \ddots & \\ & & \pi_n \end{bmatrix}$$

$$\mathbf{A}^4 = \begin{bmatrix} \pi_1 & & \\ & \ddots & \\ & & \pi_n \end{bmatrix} \begin{bmatrix} \left(p^4+q^4+6p^2q^2\right)\gamma^3 & \left(4p^3q+4pq^3\right)\gamma^3 \\ \hline \left(4p^3q+4pq^3\right)\gamma^3 & \left(p^4+q^4+6p^2q^2\right)\gamma^3 \end{bmatrix} \begin{bmatrix} \pi_1 & & \\ & \ddots & \\ & & \pi_n \end{bmatrix}$$

Note that in the above shown $\mathbf{A}^{2k}$ it is the even powers of binomial expansion of $(p+q)^{2^k}$ for $i,j$ in same class whereas it is the odd powers for $i,j$ not in the same class. We compute the filter $\mathbf{S}_{adj}$ using this fact.

$$\mathbf{S}_{adj} = \frac{1}{n}\mathbf{A}$$

$$\mathbf{S}_{adj}^k = \frac{1}{n^k}\mathbf{A}^k$$

$$\mathbf{S}_{adj}^k\mathbf{S}_{adj}^{kT} = \frac{1}{n^{2k}}\mathbf{A}^{2k}$$

$$= \begin{cases} \pi_i\pi_j\dfrac{\gamma^{2^k-1}}{n^{2k}}\displaystyle\sum_{l=0}^{2^{k-1}}\binom{2^k}{2l}p^{2^k-2l}q^{2l} & \text{if } i \text{ and } j \in \text{ same class} \\[2em] \pi_i\pi_j\dfrac{\gamma^{2^k-1}}{n^{2k}}\displaystyle\sum_{l=0}^{2^{k-1}-1}\binom{2^k}{2l+1}p^{2^k-2l-1}q^{2l+1} & \text{if } i \text{ and } j \in \text{ different class} \end{cases}$$

$$\tilde{\mathbf{\Theta}}_{adj}^{(d)} = (d+1)\mathbf{S}_{adj}^{d+1}\mathbf{S}_{adj}^{(d+1)T}$$

$$= (d+1)\pi_i\pi_j\frac{\gamma^{2^{d+1}-1}}{n^{2d+2}}\begin{cases} \displaystyle\sum_{l=0}^{2^d}\binom{2^{d+1}}{2l}p^{2^{d+1}-2l}q^{2l} & \text{if } i \text{ and } j \in \text{ same class} \\[2em] \displaystyle\sum_{l=0}^{2^d-1}\binom{2^{d+1}}{2l+1}p^{2^{d+1}-2l-1}q^{2l+1} & \text{if } i \text{ and } j \in \text{ different class} \end{cases}$$

The class separability in this case is $\zeta_{adj}^{(d)} = (d+1)c^2\frac{\gamma^{2^{d+1}-1}}{n^{2d+2}}(p-q)^{2d+2}$. The above form is not simplified as it is not an interesting case where the gap between the two blocks disappears rapidly and $\left(\tilde{\mathbf{\Theta}}_{adj}^{(\infty)}\right)_{ij} = 0$. There is no information in the kernel proving both Theorem 4 and Corollary 1. $\qquad\square$

### B.3.5 Number of Classes $K > 2$

From the above derivation for $K = 2$, it can be seen that once $\mathbf{S}_{sym}^k\mathbf{S}_{sym}^{kT}$ is computed, the population NTK for all the graph convolutions can be derived using it. Therefore, we derive it for $K > 2$ and it suffices to show the conclusions of Theorem 4 and Corollary 1. We denote the vector $\hat{\boldsymbol{\pi}}_{1k}$ with $-\pi_i\forall i \in \left[1, \frac{n}{K}\right]$, $+\pi_i\forall i \in \left[\frac{n(k-1)}{K}, \frac{nk}{K}\right]$ and $0$ for the rest. With this definition, $\mathbf{A}$ is

$$\mathbf{A} = \frac{p + (K-1)q}{K}\boldsymbol{\pi}\boldsymbol{\pi}^T + \frac{p-q}{K}\sum_{l=2}^{K}\hat{\boldsymbol{\pi}}_{1l}\hat{\boldsymbol{\pi}}_{1l}^T. \tag{29}$$

$\mathbf{D}$ for $K$ classes is $\frac{(p+(K-1)q)cn}{K}\mathrm{diag}(\boldsymbol{\pi})$ from (29). We can compute $\mathbf{S}_{sym}$ using $\mathbf{A}$ and $\mathbf{D}$ as follows,

$$
\begin{aligned}
\mathbf{S}_{sym} &= \mathbf{D}^{-\frac{1}{2}}\mathbf{A}\mathbf{D}^{-\frac{1}{2}} \\
&= \frac{K}{(p+(K-1)q)\,cn}\mathrm{diag}(\boldsymbol{\pi}^{-\frac{1}{2}})\left(\frac{p+(K-1)q}{K}\boldsymbol{\pi}\boldsymbol{\pi}^T + \frac{p-q}{K}\sum_{l=2}^{K}\hat{\boldsymbol{\pi}}_{1l}\hat{\boldsymbol{\pi}}_{1l}^T\right)\mathrm{diag}(\boldsymbol{\pi}^{-\frac{1}{2}}) \\
&= \frac{\boldsymbol{\pi}^{\frac{1}{2}}\boldsymbol{\pi}^{\frac{1}{2}T}}{cn} + \frac{p-q}{(p+(K-1)q)\,cn}\sum_{l=2}^{K}\frac{\hat{\boldsymbol{\pi}}_{1l}^{\frac{1}{2}}\hat{\boldsymbol{\pi}}_{1l}^{\frac{1}{2}T}}{cn} \\
(\mathbf{S}_{sym})_{ij} &= \frac{\sqrt{\pi_i\pi_j}}{cn}\left(1+\delta_{ij}\left(\frac{p-q}{p+(K-1)q}\right)\sum_{l=2}^{K}\frac{K}{l+l^2}\right) \\
(\mathbf{S}_{sym}^k)_{ij} &= \frac{\sqrt{\pi_i\pi_j}}{cn}\left(1+\delta_{ij}\left(\frac{p-q}{p+(K-1)q}\right)^k\sum_{l=2}^{K}\frac{K}{l+l^2}\right) \\
(\mathbf{S}_{sym}^k\mathbf{S}_{sym}^{kT})_{ij} &= \frac{\sqrt{\pi_i\pi_j}}{cn}\left(1+\delta_{ij}\left(\frac{p-q}{p+(K-1)q}\right)^{2k}\sum_{l=2}^{K}\frac{K}{l+l^2}\right)
\end{aligned}
\tag{30}
$$

It is noted that the equation (30) is very much similar to (18) for $K=2$. The further derivations of the population NTKs $\tilde{\boldsymbol{\Theta}}$ for all the convolutions are similar and the theoretical results extend without any issues. $\square$

### B.4 Corollary 3 and 4: NTK for GCN with Skip Connections

We observe that the definitions of $\mathbf{G}_i\,\forall i\in[1,d+1]$ are different for GCN with skip connections from the vanilla GCN. Despite the difference, the definition of gradient with respect to $\mathbf{W}_i$ in (7) does not change as $\mathbf{G}_i$ in the gradient accounts for the change and moreover, there is no new learnable parameter since the input transformation $\mathbf{H}_0 = \mathbf{X}\mathbf{W}_0$ where $(\mathbf{W}_0)_{ij}$ is sampled from $\mathcal{N}(0,1)$ is not learnable in our setting. Given the fact that the gradient definition holds for GCN with skip connection, the NTK will retain the form from NTK for vanilla GCN as evident from the derivation of NTK for vanilla GCN in Section B.1. The change in $\mathbf{G}_i$ will only affect the co-variance between nodes. Hence, we will derive the co-variance matrix for Skip-PC and Skip-$\alpha$ in the following.

**Skip-PC: Co-variance between nodes.** The co-variance between nodes $u$ and $v$ in $\mathbf{F}_1$ and $\mathbf{F}_i$ are derived below.

$$
\begin{aligned}
\mathbb{E}\left[(\mathbf{F}_1)_{uk}(\mathbf{F}_1)_{vk}\right] &= \langle(\mathbf{G}_1)_{u.},(\mathbf{G}_1)_{v.}\rangle \\
&= \frac{c_\sigma}{h}\langle(\mathbf{S}\sigma_s(\mathbf{H}_0))_{u.},(\mathbf{S}\sigma_s(\mathbf{H}_0))_{v.}\rangle \\
&= \frac{c_\sigma}{h}\sum_{k=1}^{h}(\mathbf{S}\sigma_s(\mathbf{H}_0))_{uk}(\mathbf{S}\sigma_s(\mathbf{H}_0))_{vk} \\
&\overset{h\to\infty}{=} c_\sigma\mathbb{E}\left[(\mathbf{S}\sigma_s(\mathbf{H}_0))_{uk}(\mathbf{S}\sigma_s(\mathbf{H}_0))_{vk}\right] \qquad ;\text{law of large numbers} \\
&= \mathbf{S}_{u.}\tilde{\mathbf{E}}_0\mathbf{S}_{.v}^T \qquad ;\tilde{\mathbf{E}}_0 = c_\sigma\underset{\mathbf{F}\sim\mathcal{N}(\mathbf{0},\mathbf{X}\mathbf{X}^T)}{\mathbb{E}}\left[\sigma_s(\mathbf{F})\sigma_s(\mathbf{F})^T\right] \\
&= (\boldsymbol{\Sigma}_1)_{uv}
\end{aligned}
\tag{31}
$$

$$\mathbb{E}\left[(\mathbf{F}_i)_{uk}\,(\mathbf{F}_i)_{vk}\right] = \langle (\mathbf{G}_i)_{u.}\,,(\mathbf{G}_i)_{v.}\rangle$$

$$= \frac{c_\sigma}{h}\,\langle (\mathbf{S}\,(\sigma(\mathbf{F}_{i-1}) + \sigma_s(\mathbf{H}_0)))_{u.}\,,(\mathbf{S}\,(\sigma(\mathbf{F}_{i-1}) + \sigma_s(\mathbf{H}_0)))_{v.}\rangle$$

$$= \frac{c_\sigma}{h}\sum_{k=1}^{h}\left(\mathbf{S}\sigma(\mathbf{F}_{i-1}) + \mathbf{S}\sigma_s(\mathbf{H}_0)\right)_{uk}\left(\mathbf{S}\sigma(\mathbf{F}_{i-1}) + \mathbf{S}\sigma_s(\mathbf{H}_0)\right)_{vk}$$

$$\overset{h\to\infty}{=} c_\sigma\mathbb{E}\left[\left(\mathbf{S}\sigma(\mathbf{F}_{i-1}) + \mathbf{S}\sigma_s(\mathbf{H}_0)\right)_{uk}\left(\mathbf{S}\sigma(\mathbf{F}_{i-1}) + \mathbf{S}\sigma_s(\mathbf{H}_0)\right)_{vk}\right]\quad;\text{law of large numbers}$$

$$= c_\sigma\Big[\mathbb{E}\left[(\mathbf{S}\sigma(\mathbf{F}_{i-1}))_{uk}\,(\mathbf{S}\sigma(\mathbf{F}_{i-1}))_{vk}\right] + \mathbb{E}\left[(\mathbf{S}\sigma(\mathbf{F}_{i-1}))_{uk}\,(\mathbf{S}\sigma_s(\mathbf{H}_0))_{vk}\right]$$

$$\quad + \mathbb{E}\left[(\mathbf{S}\sigma_s(\mathbf{H}_0))_{uk}\,(\mathbf{S}\sigma(\mathbf{F}_{i-1}))_{vk}\right] + \mathbb{E}\left[(\mathbf{S}\sigma_s(\mathbf{H}_0))_{uk}\,(\mathbf{S}\sigma_s(\mathbf{H}_0))_{vk}\right]\Big]$$

$$= \mathbf{S}_{u.}\mathbf{E}_{i-1}\mathbf{S}_{.v}^T + c_\sigma\mathbb{E}\left[(\mathbf{S}\sigma(\mathbf{F}_{i-1}))_{uk}\,(\mathbf{S}\sigma_s(\mathbf{X}\mathbf{W}_0))_{vk}\right]$$

$$\quad + c_\sigma\mathbb{E}\left[(\mathbf{S}\sigma_s(\mathbf{X}\mathbf{W}_0))_{uk}\,(\mathbf{S}\sigma(\mathbf{F}_{i-1}))_{vk}\right]$$

$$\quad + c_\sigma\mathbb{E}\left[\sum_{r=1}^{n}\sum_{s=1}^{n}\mathbf{S}_{ur}\mathbf{S}_{qs}\sigma_s\left(\mathbf{X}\mathbf{W}_0\right)_{rk}\sigma_s\left(\mathbf{X}\mathbf{W}_0\right)_{sk}\right]$$

$$\overset{(f)}{=} \mathbf{S}_{u.}\mathbf{E}_{i-1}\mathbf{S}_{.v}^T + c_\sigma\mathbf{S}_{u.}\mathbb{E}\left[\sigma_s\left(\mathbf{X}\mathbf{W}_0\right)_{rk}\sigma_s\left(\mathbf{X}\mathbf{W}_0\right)_{sk}\right]\mathbf{S}_{.v}^T$$

$$= \mathbf{S}_{u.}\mathbf{E}_{i-1}\mathbf{S}_{.v}^T + \mathbf{S}_{u.}\tilde{\mathbf{E}}_0\mathbf{S}_{.v}^T = \mathbf{S}_{u.}\mathbf{E}_{i-1}\mathbf{S}_{.v}^T + (\mathbf{\Sigma}_1)_{uv}$$

$$= (\mathbf{\Sigma}_i)_{uv} \tag{32}$$

$(f)$:  $\mathbb{E}\left[(\mathbf{S}\sigma(\mathbf{F}_{i-1}))_{uk}\,(\mathbf{S}\sigma_s(\mathbf{X}\mathbf{W}_0))_{vk}\right]$ and $\mathbb{E}\left[(\mathbf{S}\sigma_s(\mathbf{X}\mathbf{W}_0))_{uk}\,(\mathbf{S}\sigma(\mathbf{F}_{i-1}))_{vk}\right]$ evaluate to 0 by conditioning on $\mathbf{W}_0$ first and rewriting the expectation based on this conditioning. The terms within expectation are independent when conditioned on $\mathbf{W}_0$, and hence it is

$\underset{\mathbf{W}_0}{\mathbb{E}}\left[\underset{\mathbf{\Sigma}_{i-1}|\mathbf{W}_0}{\mathbb{E}}\left[(\mathbf{S}\sigma(\mathbf{F}_{i-1}))_{uk}\,|\mathbf{W}_0\right]\underset{\mathbf{\Sigma}_{i-1}|\mathbf{W}_0}{\mathbb{E}}\left[(\mathbf{S}\sigma_s(\mathbf{X}\mathbf{W}_0))_{vk}\,|\mathbf{W}_0\right]\right]$ by taking $h$ in $\mathbf{W}_0$ going to infinity first. Here, $\underset{\mathbf{\Sigma}_{i-1}|\mathbf{W}_0}{\mathbb{E}}\left[(\mathbf{S}\sigma_s(\mathbf{X}\mathbf{W}_0))_{vk}\,|\mathbf{W}_0\right] = 0$.

We get the co-variance matrix for all pairs of nodes $\mathbf{\Sigma}_1 = \mathbf{S}\tilde{\mathbf{E}}_0\mathbf{S}^T$ and $\mathbf{\Sigma}_i = \mathbf{S}\mathbf{E}_{i-1}\mathbf{S}^T + \mathbf{\Sigma}_1$ from (31) and (32).

**Skip-$\alpha$: Co-variance between nodes.** Let $u$ and $v$ be two nodes and the co-variance between $u$ and $v$ in $\mathbf{F}_1$ and $\mathbf{F}_i$ are derived below.

$$\mathbb{E}\left[(\mathbf{F}_1)_{uk}\,(\mathbf{F}_1)_{vk}\right] = \langle (\mathbf{G}_1)_{u.}\,,(\mathbf{G}_1)_{v.}\rangle$$

$$= \frac{c_\sigma}{h}\sum_{k=1}^{h}\left((1-\alpha)\mathbf{S}\sigma_s(\mathbf{H}_0) + \alpha\sigma_s(\mathbf{H}_0)\right)_{uk}\left((1-\alpha)\mathbf{S}\sigma_s(\mathbf{H}_0) + \alpha\sigma_s(\mathbf{H}_0)\right)_{vk}$$

$$\overset{h\to\infty}{=} c_\sigma\mathbb{E}\left[\left((1-\alpha)\mathbf{S}\sigma_s(\mathbf{H}_0) + \alpha\sigma_s(\mathbf{H}_0)\right)_{uk}\left((1-\alpha)\mathbf{S}\sigma_s(\mathbf{H}_0) + \alpha\sigma_s(\mathbf{H}_0)\right)_{vk}\right]$$

$$= c_\sigma\Big[(1-\alpha)^2\mathbb{E}\left[(\mathbf{S}\sigma_s(\mathbf{H}_0))_{uk}\,(\mathbf{S}\sigma_s(\mathbf{H}_0))_{vk}\right]$$

$$\quad + (1-\alpha)\alpha\left(\mathbb{E}\left[(\mathbf{S}\sigma_s(\mathbf{H}_0))_{uk}\,(\sigma_s(\mathbf{H}_0))_{vk}\right] + \mathbb{E}\left[(\mathbf{S}\sigma_s(\mathbf{H}_0))_{vk}\,(\sigma_s(\mathbf{H}_0))_{uk}\right]\right)$$

$$\quad + \alpha^2\mathbb{E}\left[(\sigma_s(\mathbf{H}_0))_{uk}\,(\sigma_s(\mathbf{H}_0))_{vk}\right]\Big]$$

$$= (1-\alpha)^2\mathbf{S}_{u.}\tilde{\mathbf{E}}_0\mathbf{S}_{.v}^T + (1-\alpha)\alpha\left(\mathbf{S}_{u.}\left(\tilde{\mathbf{E}}_0\right)_{.v} + \left(\tilde{\mathbf{E}}_0\right)_{u.}\mathbf{S}_{.v}^T\right) + \alpha^2\left(\tilde{\mathbf{E}}_0\right)_{uv}$$

$$= (\mathbf{\Sigma}_1)_{uv} \tag{33}$$

Using $\mathbb{E}\left[(\mathbf{F}_1)_{uk}\,(\mathbf{F}_1)_{vk}\right]$, we recursively evalaue $\mathbb{E}\left[(\mathbf{F}_i)_{uk}\,(\mathbf{F}_i)_{vk}\right]$ in the following,

$$\mathbb{E}\left[(\mathbf{F}_i)_{uk}(\mathbf{F}_i)_{vk}\right] = \langle (\mathbf{G}_i)_{u.}, (\mathbf{G}_i)_{v.} \rangle$$

$$= \frac{c_\sigma}{h} \sum_{k=1}^{h} \left((1-\alpha)\mathbf{S}\sigma(\mathbf{F}_{i-1}) + \alpha\sigma_s(\mathbf{H}_0)\right)_{uk} \left((1-\alpha)\mathbf{S}\sigma(\mathbf{F}_{i-1}) + \alpha\sigma_s(\mathbf{H}_0)\right)_{vk}$$

$$\overset{h\to\infty}{=} c_\sigma \mathbb{E}\left[\left((1-\alpha)\mathbf{S}\sigma(\mathbf{F}_{i-1}) + \alpha\sigma_s(\mathbf{H}_0)\right)_{uk} \left((1-\alpha)\mathbf{S}\sigma(\mathbf{F}_{i-1}) + \alpha\sigma_s(\mathbf{H}_0)\right)_{vk}\right]$$

$$= c_\sigma \Big[ (1-\alpha)^2 \mathbb{E}\left[(\mathbf{S}\sigma(\mathbf{F}_{i-1}))_{uk} (\mathbf{S}\sigma(\mathbf{F}_{i-1}))_{vk}\right] + \alpha^2 \mathbb{E}\left[(\sigma_s(\mathbf{H}_0))_{uk} (\sigma_s(\mathbf{H}_0))_{vk}\right]$$

$$+ (1-\alpha)\alpha \left( \mathbb{E}\left[(\mathbf{S}\sigma(\mathbf{F}_{i-1}))_{uk} (\sigma_s(\mathbf{H}_0))_{vk}\right] + \mathbb{E}\left[(\sigma_s(\mathbf{H}_0))_{uk} (\mathbf{S}\sigma(\mathbf{F}_{i-1}))_{vk}\right] \right) \Big]$$

$$\overset{(g)}{=} (1-\alpha)^2 \mathbf{S}_{u.}\mathbf{E}_{i-1}\mathbf{S}_{.v}^T + \alpha^2 \left(\tilde{\mathbf{E}}_0\right)_{uv} = (\Sigma_i)_{uv} \tag{34}$$

$(g)$: same argument as $(f)$ in derivation of $\Sigma_i$ in Skip-PC.

We get the co-variance matrix for all pairs of nodes $\Sigma_1 = (1-\alpha)^2 \mathbf{S}\tilde{\mathbf{E}}_0\mathbf{S}^T + \alpha(1-\alpha)\left(\mathbf{S}\tilde{\mathbf{E}}_0 + \tilde{\mathbf{E}}_0\mathbf{S}^T\right) + \alpha^2\tilde{\mathbf{E}}_0$ and $\Sigma_i = (1-\alpha)^2 \mathbf{S}\mathbf{E}_{i-1}\mathbf{S}^T + \alpha^2\tilde{\mathbf{E}}_0$ from (33) and (34).

## B.5  Theorem 5: Class Separability of Population NTK $\tilde{\Theta}$ for Skip-PC

NTK at depth $d$, $\Theta_{PC}^{(d)}$ for Skip-PC with linear activations is

$$\Theta_{PC}^{(d)} = \sum_{k=1}^{d+1} \mathbf{S}^{d+1-k} \Sigma_k \mathbf{S}^{(d+1-k)T}$$

$$= \sum_{k=1}^{d+1} \mathbf{S}^{d+1-k} \left(\mathbf{S}^k \mathbf{S}^{kT} + \mathbf{S}\mathbf{S}^T\right) \mathbf{S}^{(d+1-k)T}$$

$$= \sum_{k=1}^{d+1} \underbrace{\mathbf{S}^{d+1}\mathbf{S}^{(d+1)T}}_{I} + \underbrace{\mathbf{S}^{d+2-k}\mathbf{S}^{(d+2-k)T}}_{II} \tag{35}$$

In (35), $I$ is NTK without skip connection and $II$ is computed for $\mathbf{S}_{row}$ and $\mathbf{S}_{sym}$ as follows.

Computing $II$ for population NTK $\tilde{\Theta}^{(d)}$ for $\mathbf{S}_{sym}$: for nodes $i$ and $j$,

$$\sum_{k=1}^{d+1} \left(\mathbf{S}_{sym}^{d+2-k}\mathbf{S}_{sym}^{(d+2-k)T}\right)_{ij} = \sum_{k=1}^{d+1} \left(1 + \delta_{ij} r^{2d+4-2k}\right) \sqrt{\pi_i \pi_j}\, (cn)^{-1}$$

$$= (d+1)\frac{\sqrt{\pi_i \pi_j}}{cn} + \delta_{ij}\frac{\sqrt{\pi_i \pi_j}}{cn} \sum_{k=1}^{d+1} r^{2k}$$

$$= (d+1)\frac{\sqrt{\pi_i \pi_j}}{cn} + \delta_{ij}\frac{\sqrt{\pi_i \pi_j}}{cn} \frac{r^2\left(1 - r^{2(d+1)}\right)}{1 - r^2} \tag{36}$$

Combining (36) with (19), the class separability of the kernel $\zeta_{PC,sym}^{(d)}$ as $d \to \infty$ is determined only by the last term in (36) as the other terms give 0 separation. Hence, the influence of skip connection gives

$$\zeta_{PC,sym}^{(\infty)} = \frac{16\tau^2 r^2}{n^2 (cn)(1 - r^2)} \tag{37}$$

where $\tau$ is defined as in Theorem 4.

$\sqrt{\pi_i \pi_j}\left(2 + \delta_{ij} r^2\right)$. Thus showing class separation information retained even at $\infty$ depth and graph size. $\quad\square$

Similarly, computing $II$ for $\mathbf{S}_{row}$ without assumption on $\gamma$, $i$ and $j$ in class 1,

$$\sum_{k=1}^{d+1} \left( \mathbf{S}_{row}^{d+2-k} \mathbf{S}_{row}^{(d+2-k)T} \right)_{ij} = (cn)^{-2} \sum_{k=1}^{d+1} \left( 1 + r^{2k} + 2r^k \right) \lambda + \left( 1 + r^{2k} - 2r^k \right) \mu$$

$$= (cn)^{-2} \left( (\lambda + \mu) \left( (d+1) + \frac{r^2 \left( 1 - r^{2(d+1)} \right)}{1 - r^2} \right) + 2(\lambda - \mu) \frac{r \left( 1 - r^{d+1} \right)}{1 - r} \right) \quad (38)$$

For $i$ and $j$ in class 2,

$$\sum_{k=1}^{d+1} \left( \mathbf{S}_{row}^{d+2-k} \mathbf{S}_{row}^{(d+2-k)T} \right)_{ij} = (cn)^{-2} \sum_{k=1}^{d+1} \left( 1 + r^{2k} - 2r^k \right) \lambda + \left( 1 + r^{2k} + 2r^k \right) \mu$$

$$= (cn)^{-2} \left( (\lambda + \mu) \left( (d+1) + \frac{r^2 \left( 1 - r^{2(d+1)} \right)}{1 - r^2} \right) + 2(-\lambda + \mu) \frac{r \left( 1 - r^{d+1} \right)}{1 - r} \right)$$
$$(39)$$

For $i$ and $j$ in different class,

$$\sum_{k=1}^{d+1} \left( \mathbf{S}_{row}^{d+2-k} \mathbf{S}_{row}^{(d+2-k)T} \right)_{ij} = (cn)^{-2} \sum_{k=1}^{d+1} \left( 1 - r^{2k} \right) (\lambda + \mu)$$

$$= (cn)^{-2} (\lambda + \mu) \left( (d+1) - \frac{r^2 \left( 1 - r^{2(d+1)} \right)}{1 - r^2} \right) \quad (40)$$

Therefore, the influence of the skip connection in the class separability of population NTK $\tilde{\mathbf{\Theta}}_{PC,row}^{(\infty)}$ with $\gamma$ assumption is obtained by substituting $\lambda + \mu = 2\gamma$ and $\lambda - \mu = 0$ in (38), (39) and (40) .

$$\zeta_{PC,row}^{(\infty)} = \frac{8\gamma r^2}{(cn)^2 (1 - r^2)}$$

hence deriving Theorem 5. $\qquad\square$

## B.6 Theorem 6: Population NTK $\tilde{\mathbf{\Theta}}$ for Skip-$\alpha$

We expand $\mathbf{\Sigma}_1$ and $\mathbf{\Sigma}_k$ of Skip-$\alpha$ first to derive the population NTK.

$$\mathbf{\Sigma_1} = (1 - \alpha)^2 \mathbf{S}\mathbf{S}^T + \alpha (1 - \alpha) \left( \mathbf{S} + \mathbf{S}^T \right) + \alpha^2 \mathbf{I}_n$$

$$\mathbf{\Sigma_k} = (1 - \alpha)^2 \mathbf{S}\mathbf{\Sigma_{k-1}}\mathbf{S^T} + \alpha^2 \mathbf{I_n}$$

$$= (1 - \alpha)^{2k} \mathbf{S}^k \mathbf{S}^{kT} + \alpha (1 - \alpha)^{2k-1} \mathbf{S}^{k-1} \left( \mathbf{S} + \mathbf{S}^T \right) \mathbf{S}^{k-1^T} + \alpha^2 \sum_{l=0}^{k-1} (1 - \alpha)^{2l} \mathbf{S}^l \mathbf{S}^{lT} \quad (41)$$

Exact NTK of depth $d$ for Skip-$\alpha$ is expanded using the above as follows.

$$\mathbf{\Theta}_\alpha^{(d)} = \sum_{k=1}^{d+1} \mathbf{S}^{d+1-k} \mathbf{\Sigma}_k \mathbf{S}^{(d+1-k)T}$$

$$= \sum_{k=1}^{d+1} \underbrace{(1 - \alpha)^{2k} \mathbf{S}^{d+1} \mathbf{S}^{(d+1)T}}_{I} + \underbrace{\alpha (1 - \alpha)^{2k-1} \mathbf{S}^d \left( \mathbf{S} + \mathbf{S}^T \right) \mathbf{S}^{d^T}}_{II} + \underbrace{\alpha^2 \sum_{l=0}^{k-1} (1 - \alpha)^{2l} \mathbf{S}^{d+1-k+l} \mathbf{S}^{(d+1-k+l)T}}_{III}$$
$$(42)$$

We compute the class separability of the kernel $\mathbf{\Theta}_\alpha^{(\infty)}$ as $d \to \infty$ for $\mathbf{S}_{sym}$ and $\mathbf{S}_{row}$. From (42), it is clear that terms $I$ and $II$ lead to 0 class separation as derived in previous cases. So, we evaluate $III$ of (42) in the following.

$$III_{ij} = \alpha^2 \sum_{k=1}^{d+1} \sum_{l=0}^{k-1} (1-\alpha)^{2l} \, \mathbf{S}_{sym}^{d+1-k+l} \mathbf{S}_{sym}^{(d+1-k+l)T}$$

$$= \frac{\sqrt{\pi_i \pi_j}}{cn} \alpha^2 \sum_{k=1}^{d+1} \sum_{l=0}^{k-1} (1-\alpha)^{2l} \left( 1 + \delta_{ij} r^{2d+2-2k+2l} \right)$$

$$= \frac{\sqrt{\pi_i \pi_j}}{cn} \alpha^2 \sum_{k=1}^{d+1} \frac{1 - (1-\alpha)^{2k}}{1 - (1-\alpha)^2} + \delta_{ij} \frac{r^{2(d+1-k)} \left( 1 - \left( (1-\alpha)^2 r^2 \right)^k \right)}{1 - (1-\alpha)^2 r^2}$$

$$= \frac{\sqrt{\pi_i \pi_j} \alpha^2}{cn} \left[ \frac{(d+1)}{1 - (1-\alpha)^2} - \frac{(1-\alpha)^2 \left( 1 - (1-\alpha)^{2(d+1)} \right)}{\left( 1 - (1-\alpha)^2 \right)^2} + \right.$$

$$\left. \frac{\delta_{ij}}{1 - (1-\alpha)^2 r^2} \left( \frac{1 - r^{2(d+1)}}{1 - r^2} - r^{2(d+1)} \frac{(1-\alpha)^2 \left( 1 - (1-\alpha)^{2(d+1)} \right)}{1 - (1-\alpha)^2} \right) \right] \qquad (43)$$

The class separability of kernel is non zero only for the last term in (43). Hence, the class separability $\zeta_{\alpha,sym}^{(d)}$ is

$$\zeta_{\alpha,sym}^{(d)} = \frac{16\tau^2 \alpha^2}{(cn)n^2 \left( 1 - (1-\alpha)^2 r^2 \right)} \left( \frac{1 - r^{2(d+1)}}{1 - r^2} - r^{2(d+1)} \frac{(1-\alpha)^2 \left( 1 - (1-\alpha)^{2(d+1)} \right)}{1 - (1-\alpha)^2} \right)$$

$$\zeta_{\alpha,sym}^{(\infty)} = \frac{16\tau^2 \alpha^2}{(cn)n^2 \left( 1 - (1-\alpha)^2 r^2 \right)} \left( \frac{1}{1 - r^2} \right)$$

proving Theorem 6. $\qquad \square$

We now compute $III$ for population NTK $\tilde{\Theta}_\alpha^{(\infty)}$ using $\mathbf{S}_{row}$ under $\lambda = \mu = \gamma$. The derivation holds without this consideration as well.

$$III_{ij} = \alpha^2 \sum_{k=1}^{d+1} \sum_{l=0}^{k-1} (1-\alpha)^{2l} \, \mathbf{S}_{row}^{d+1-k+l} \mathbf{S}_{row}^{(d+1-k+l)T}$$

$$= \frac{2\gamma}{(cn)^2} \alpha^2 \sum_{k=1}^{d+1} \sum_{l=0}^{k-1} (1-\alpha)^{2l} \left( 1 + \delta_{ij} r^{2d+2-2k+2l} \right)$$

$$= \frac{2\gamma}{(cn)^2} \alpha^2 \sum_{k=1}^{d+1} \frac{1 - (1-\alpha)^{2k}}{1 - (1-\alpha)^2} + \delta_{ij} \frac{r^{2(d+1-k)} \left( 1 - \left( (1-\alpha)^2 r^2 \right)^k \right)}{1 - (1-\alpha)^2 r^2}$$

$$= \frac{2\gamma \alpha^2}{(cn)^2} \left[ \frac{(d+1)}{1 - (1-\alpha)^2} - \frac{(1-\alpha)^2 \left( 1 - (1-\alpha)^{2(d+1)} \right)}{\left( 1 - (1-\alpha)^2 \right)^2} + \right.$$

$$\left. \frac{\delta_{ij}}{1 - (1-\alpha)^2 r^2} \left( \frac{1 - r^{2(d+1)}}{1 - r^2} - r^{2(d+1)} \frac{(1-\alpha)^2 \left( 1 - (1-\alpha)^{2(d+1)} \right)}{1 - (1-\alpha)^2} \right) \right] \qquad (44)$$

Similar to $\mathbf{S}_{sym}$, the class separability of kernel is non zero only for the last term in (44). Hence, the class separability $\zeta_{\alpha,row}^{(d)}$ is

$$\zeta_{\alpha,row}^{(d)} = \frac{8\gamma\alpha^2}{(cn)^2\left(1-(1-\alpha)^2 r^2\right)}\left(\frac{1-r^{2(d+1)}}{1-r^2}-r^{2(d+1)}\frac{(1-\alpha)^2\left(1-(1-\alpha)^{2(d+1)}\right)}{1-(1-\alpha)^2}\right)$$

$$\zeta_{\alpha,row}^{(\infty)} = \frac{8\gamma\alpha^2}{(cn)^2\left(1-(1-\alpha)^2 r^2\right)}\left(\frac{1}{1-r^2}\right)$$

proving Theorem 6. $\qquad\square$

## B.7 Theorem 3: Population NTK $\tilde{\Theta}$ for ReLU GCN for normalized adjacency S

We first state the NTK for ReLU GCN using the general NTK Theorem 1 and result from Bietti & Mairal (2019) in the following corollary. Note that $c_\sigma = 2$ for ReLU activation.

**Corollary 5 (ReLU GCN)** *Consider* $\sigma(x) := ReLU(x)$ *in* $F_{\mathbf{W}}(\mathbf{X}, \mathbf{S})$. *The NTK is computed as in* (3), *where given* $\mathbf{\Sigma}_k$ *at each layer, one can evaluate the entries of* $\mathbf{E}_k$ *and* $\dot{\mathbf{E}}_k$ *using a result from Bietti & Mairal* *(2019) as*

$$\left(\mathbf{E}_k\right)_{ij} = \sqrt{(\mathbf{\Sigma}_k)_{ii}(\mathbf{\Sigma}_k)_{jj}}\;\kappa_1\left(\frac{(\mathbf{\Sigma}_k)_{ij}}{\sqrt{(\mathbf{\Sigma}_k)_{ii}(\mathbf{\Sigma}_k)_{jj}}}\right)$$

$$\left(\dot{\mathbf{E}}_k\right)_{ij} = \kappa_0\left(\frac{(\mathbf{\Sigma}_k)_{ij}}{\sqrt{(\mathbf{\Sigma}_k)_{ii}(\mathbf{\Sigma}_k)_{jj}}}\right),$$

where $\kappa_0(x) = \frac{1}{\pi}\left(\pi - \arccos(x)\right)$ and $\kappa_1(x) = \frac{1}{\pi}\left(x\left(\pi - \arccos(x)\right) + \sqrt{1-x^2}\right)$.

Using Corollary 5, we derive Theorem 3, the population NTK of the ReLU GCN for depth $d$, $\tilde{\Theta}_{ReLU}^{(d)}$ considering homogeneous degree correction $\boldsymbol{\pi}$. That is, $\boldsymbol{\pi} = (c, \ldots, c)^T$. Therefore, symmetric, row and column normalized adjacencies are equivalent and is,

$$\mathbf{S} = \mathbf{D}^{-\frac{1}{2}}\mathbf{A}\mathbf{D}^{-\frac{1}{2}}$$

$$= \frac{1}{n}\left(\mathbf{1}\mathbf{1}^T + r\hat{\mathbf{1}}\hat{\mathbf{1}}^T\right)$$

Therefore, using $\mathbf{S}$, $\kappa_0(.)$ and $\kappa_1(.)$ we compute $\mathbf{\Sigma}_1$, $\mathbf{E}_1$ and $\dot{\mathbf{E}}_1$ as,

$$\mathbf{\Sigma}_1 = \mathbf{S}\mathbf{S}^T = \frac{1}{n}\left[\begin{array}{c|c}\dfrac{1+r^2}{1-r^2} & \dfrac{1-r^2}{1+r^2}\end{array}\right]_{n\times n}$$

$$\mathbf{E}_1 = \frac{1}{n}\left(1+r^2\right)\left[\begin{array}{c|c}1 & \kappa_1\left(\frac{1-r^2}{1+r^2}\right) \\ \hline \kappa_1\left(\frac{1-r^2}{1+r^2}\right) & 1\end{array}\right]_{n\times n}$$

$$= \frac{1}{n}\left(1+r^2\right)\left[\begin{array}{c|c}1 & \kappa_1\left(\Delta_1\right) \\ \hline \kappa_1\left(\Delta_1\right) & 1\end{array}\right]_{n\times n}\quad;\Delta_1 := \frac{1-r^2}{1+r^2}$$

$$\dot{\mathbf{E}}_1 = \left[\begin{array}{c|c}1 & \kappa_0\left(\Delta_1\right) \\ \hline \kappa_0\left(\Delta_1\right) & 1\end{array}\right]_{n\times n}\tag{45}$$

Now, lets define $\Delta_k := \dfrac{(1-r^2)+(1+r^2)\kappa_1(\Delta_{k-1})}{(1+r^2)+(1-r^2)\kappa_1(\Delta_{k-1})}$. Furthermore, $\Delta_k^n$ and $\Delta_k^d$ denote the numerator and denominator of $\Delta_k$, respectively. With this definition, we compute $\boldsymbol{\Sigma}_k$, $\mathbf{E}_k$ and $\dot{\mathbf{E}}_k$ recursive as follows to compute the population NTK $\tilde{\boldsymbol{\Theta}}^{(d)}$,

$$\boldsymbol{\Sigma}_2 = \mathbf{S}\mathbf{E}_1\mathbf{S}^T = \frac{\Delta_1^d}{2n}\left[\begin{array}{c|c}\Delta_2^d & \Delta_2^n \\ \hline \Delta_2^n & \Delta_2^d\end{array}\right]_{n\times n}$$

$$\mathbf{E}_2 = \frac{\Delta_1^d\Delta_2^d}{2n}\left[\begin{array}{c|c}1 & \kappa_1(\Delta_2) \\ \hline \kappa_1(\Delta_2) & 1\end{array}\right]_{n\times n} \; ; \quad \dot{\mathbf{E}}_2 = \left[\begin{array}{c|c}1 & \kappa_0(\Delta_2) \\ \hline \kappa_0(\Delta_2) & 1\end{array}\right]_{n\times n}$$

Extending to $k$,

$$\boldsymbol{\Sigma}_k = \frac{\Delta_1^d\cdots\Delta_{k-1}^d}{2^{k-1}n}\left[\begin{array}{c|c}\Delta_k^d & \Delta_k^n \\ \hline \Delta_k^n & \Delta_k^d\end{array}\right]_{n\times n}$$

$$\mathbf{E}_k = \frac{\Delta_1^d\cdots\Delta_k^d}{2^{k-1}n}\left[\begin{array}{c|c}1 & \kappa_1(\Delta_k) \\ \hline \kappa_1(\Delta_k) & 1\end{array}\right]_{n\times n} \; ; \quad \dot{\mathbf{E}}_k = \left[\begin{array}{c|c}1 & \kappa_0(\Delta_k) \\ \hline \kappa_0(\Delta_k) & 1\end{array}\right]_{n\times n} \tag{46}$$

We obtain population NTK for ReLU GCN in Theorem 3 by substituting $\boldsymbol{\Sigma}_k$, $\boldsymbol{\Sigma}_1$ and $\dot{\mathbf{E}}_k$ in the NTK equation in (3). $\qquad\square$

## B.8 Difference between block difference of linear and ReLU GCNs for depth $d = 1$

First, lets compute the average in-class and out-of-class block differences for $d = 1$ linear and ReLU GCNs. To do so, lets consider homogeneous degree correction as in Section B.7. Therefore, population NTKs for linear and ReLU GCNs $\tilde{\boldsymbol{\Theta}}^{(1)}$ and $\tilde{\boldsymbol{\Theta}}_{ReLU}^{(1)}$ are,

$$\tilde{\boldsymbol{\Theta}}^{(1)} = \frac{2}{n}\left[\begin{array}{c|c}1+r^2 & 1-r^2 \\ \hline 1-r^2 & 1+r^2\end{array}\right]_{n\times n} \tag{47}$$

$$\tilde{\boldsymbol{\Theta}}_{ReLU}^{(1)} = \frac{1}{2n}\left[\begin{array}{c|c}\left(1+r^2\right)^2+\left(1-r^2\right)^2\kappa_0(\Delta_1) & \left(1-r^4\right)+\left(1-r^4\right)\kappa_0(\Delta_1) \\ \hline \left(1-r^4\right)+\left(1-r^4\right)\kappa_0(\Delta_1) & \left(1+r^2\right)^2+\left(1-r^2\right)^2\kappa_0(\Delta_1)\end{array}\right]_{n\times n} + \frac{\Delta_1^d}{2n}\left[\begin{array}{c|c}\Delta_2^d & \Delta_2^n \\ \hline \Delta_2^n & \Delta_2^d\end{array}\right]_{n\times n} \tag{48}$$

Let the average block difference for linear and ReLU GCNs of depth 1 be denoted by $\zeta_{lin}$ and $\zeta_{ReLU}$, respectively. Using (47) and (48), we get

$$\zeta_{lin} = \frac{8r^2}{n} = \mathcal{O}\left(\frac{r^2}{n}\right)$$

$$\zeta_{ReLU} = \frac{4r^2\left(r^2+1+\left(r^2-1\right)\kappa_0(\Delta_1)\right)}{2n} + \frac{4r^2\left(1+r^2\right)\left(1-\kappa_1(\Delta_1)\right)}{2n}$$

$$= \mathcal{O}\left(\frac{r^2}{n}\right)$$

Therefore, theoretically linear GCN and ReLU GCN of depth 1 retains similar class information for large graphs and hence they perform similarly. $\qquad\square$

## B.9 Analysis without orthonormal feature assumption $\mathbf{X}\mathbf{X}^T \neq \mathbf{I}_n$

To include the features so that $\mathbf{X}\mathbf{X}^T \neq \mathbf{I}_n$, we consider *Contextual Stochastic Block Models* (Deshpande et al., 2018) in which the features of node $i$, $\mathbf{x}_i \sim z_i\boldsymbol{\mu} + \mathcal{N}(0,\sigma^2\mathbf{I}_f)$, where $\boldsymbol{\mu} \in \mathbb{R}^f$ and $z_i = +1$ if node $i \in \mathcal{C}_1$, $-1$ if $i \in \mathcal{C}_2$ for $K = 2$. The analysis can be extended to $K > 2$ as well. Under this model, the

population version of $\mathbf{X}\mathbf{X}^T$ is $\mathbf{z}\boldsymbol{\mu}^T\boldsymbol{\mu}\mathbf{z}^T = ||\boldsymbol{\mu}||^2\mathbf{z}\mathbf{z}^T$ where $\mathbf{z} = (z_1, \ldots, z_n) \in \mathbb{R}^n$. For simplicity, we present the average in-class and out-of-class block difference of linear ($\zeta_{lin}$) and ReLU GCNs ($\zeta_{ReLU}$) for depth $d = 1$. $\zeta_{lin} = ||\boldsymbol{\mu}||^2\left(2r^4\right)$ and $\zeta_{ReLU} = ||\boldsymbol{\mu}||^2\left(4r^4(1 + 1/n)\right)$, respectively. Consequently, $\zeta_{lin} \leq \zeta_{ReLU} \forall r \in [0,1], n$. However, both are of $\mathcal{O}(r^4)$. As the population NTK for depth $d$ will be a more complex expression under Contextual SBM, we show the result for $d = 1$ for simplicity. But, we note that the result will extend to general $d$. $\qquad\square$

# C   Empirical Analysis

We provide the code for NTK and the block model in
`https://github.com/mahalakshmi-sabanayagam/NTK_GCN`.

## C.1   Experimental Details of Figure 1

We use the code for GCN without skip connections from github1(Kipf & Welling, 2017) and skip connection from github2(Chen et al., 2020). The following hyperparameters are used for GCN without skip connections: learning rate is 0.01, weight decay is $5e-4$, hidden layer width is 64 and epochs is $500, 1500, 2000$ for depths $2, 4, 8$ respectively. For the skip connections, we used GCNII model, same parameters as vanilla GCN with $\alpha = 0.1$. The performance is averaged over 5 runs.

In Figure 8, we showcase the performance degradation of GCN with depth. The right plot shows the zoomed in version of the left plot to show the performance drop more clearly. Note that depth refers to the number of hidden layers in the definition of GCN (2). Hence, depth$= 0$ means there is no hidden layer.

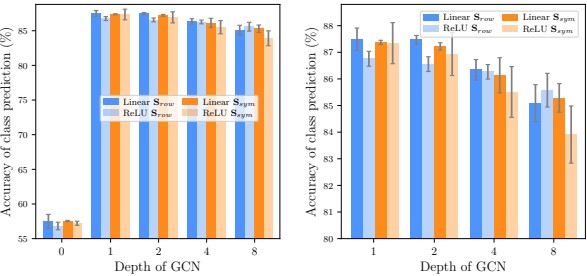

Figure 8: **Performance of GCN with depth on Cora.** Depth$= 0$ refers to no hidden layer in GCN. The right plot shows the zoomed in version of the left plot.

## C.2   Comparison of GCN and NTK

Although it is theoretically clear that the infinite width assumption should not affect the observations made on performance of GCN with $\mathbf{S}_{sym}$ and $\mathbf{S}_{row}$ in Figure 1, we illustrate the same using graph NTK. Figure 9 shows that the observation is seen in graph NTK as well, thus supporting our theoretical argument.

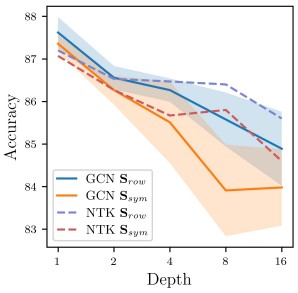

Figure 9: **Comparison of the accuracy of a trained finite width GCN and the corresponding NTK.** NTK captures the performance trend of the GCN, although the exact performance doesn't match.

### C.3 Numerical Validation for DC-SBM for Vanilla GCN and Skip-$\alpha$

**Experimental Details**. For the experiments, we fix the size of the sampled graphs to $n = 1000$, $p = 0.8$ and $q = 0.1$ for homophily DC-SBM, $p = 0.1$ and $q = 0.8$ for heterophily DC-SBM and $p = q = 1$ for core-periphery DC-SBM. $\boldsymbol{\pi}$ is sampled uniformly $[0, 1]$ for homophily and heterophily, and $\pi_i \sim \text{Unif}(0.5, 1) \forall i \in core$ and $\pi_i \sim \text{Unif}(0, 0.5) \forall i \in periphery$ for core-periphery DC-SBM.

**Illustration of impact of depth in Vanilla GCN using Homophily DC-SBM**. We show the impact of depth in Vanilla GCN using homophily DC-SBM in Figure 10. The DC-SBM is shown in the first column and columns 2 and 3 show the exact NTK for depth=1 and 8 for symmetric and row normalization, respectively. The plots clearly illustrate the complete loss of class information in symmetric normalization with depth (column 2). While the prevalence of block difference has decresed in row normalization over depth (column 3), the block/community structure is still retained. Thus showing the strong representation power of $\mathbf{S}_{row}$.

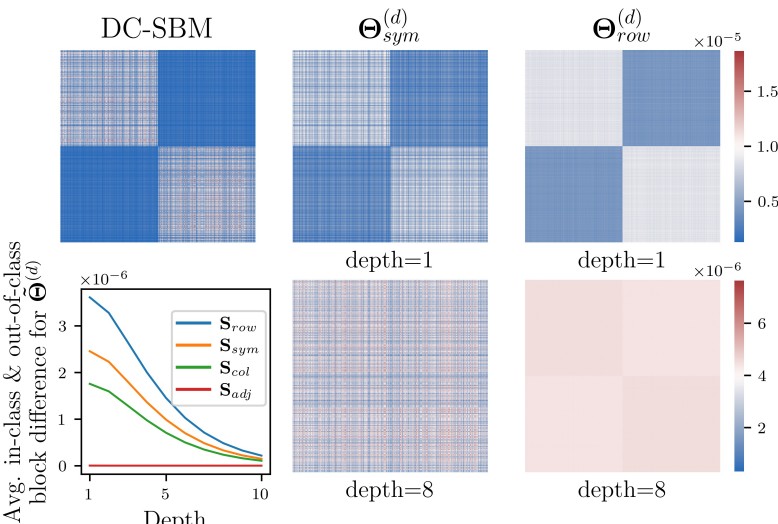

Figure 10: **Numerical validation of Theorem 4 using DC-SBM** shown in the first plot of column 1. Columns 2 and 3 illustrate the exact NTKs of depth=1 and 8 for $\mathbf{S}_{sym}$ and $\mathbf{S}_{row}$, respectively. Second plot in column 1 shows the average gap between in-class and out-of-class blocks from theory.

**Illustration of $\mathbf{S}_{col}$ and $\mathbf{S}_{adj}$ in Vanilla GCN using Homophily DC-SBM**. We extend the experiments on numerical validation for random graphs using vanilla GCN described in Section 5.2 to column normalized adjacency $\mathbf{S}_{col}$ and unnormalized adjacency $\mathbf{S}_{adj}$ here. We use the same setup described in Section 5.2 and Figure 11 illustrates the results. We observe that even for depth 1 both the convolutions are influenced by

the degree correction and there is no class information in the kernels for higher depth. Thus, this validates the theoretical result in Theorem 4.

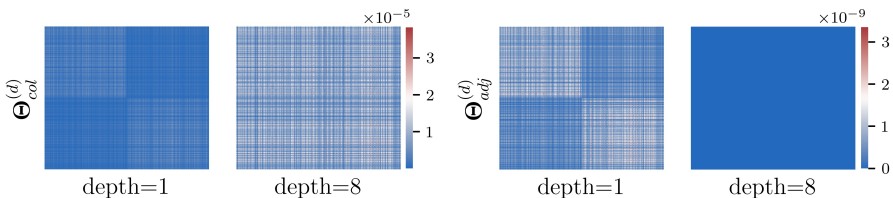

Figure 11: **Numerical validation of DC-SBM for Vanilla GCN.** The first two heatmaps show the exact NTK $\boldsymbol{\Theta}^{(d)}$ for column normalized adjacency convolution $\mathbf{S}_{col}$ and the other two for unnormalized adjacency $\mathbf{S}_{adj}$ for depths $d = 1$ and $8$.

**Validation of the theoretical filter ordering based on the population kernel block difference**. We validate the theoretical finding of the filter $\tilde{\boldsymbol{\Theta}}_{row} \succ \tilde{\boldsymbol{\Theta}}_{sym} \succ \tilde{\boldsymbol{\Theta}}_{col} \succ \tilde{\boldsymbol{\Theta}}_{adj}$ based on the population kernel block difference by sampling a graph from a DC-SBM and measuring the Mean Squared Error (MSE) of the prediction from the exact kernel for various depth of GCN. Figure 12 illustrates the order of convolution filters obtained theoretically holds very well in practice.

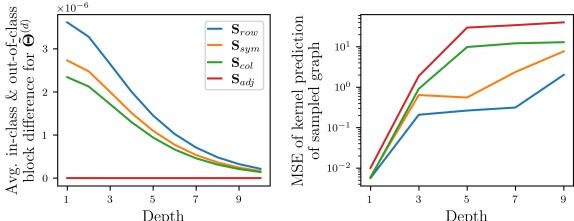

Figure 12: **Numerical validation of the theoretical filter ordering based on the kernel class separability.** Left plot shows the result from theory based on the block difference of the population NTK. The right plot shows the Mean Squared Error (MSE) of the prediction from the exact kernel of a sampled graph. The order of convolutions based on MSE clearly validates the theory.

**Illustration of impact of depth in Skip-PC and Skip-$\alpha$ using Homophily DC-SBM**. We present a complementary result to Section 6.3 here. We use the same setting as described in Section 6.3 and plot the exact NTKs of depths 1 and 8 for symmetric and row normalization. Figure 13 shows the results for Skip-PC and we observe that the gap between in-class and out-of-class blocks decreases for both $\mathbf{S}_{row}$ and $\mathbf{S}_{sym}$ with depth, but the class information is still retained for larger depth and the gap doesn't vanish. Between $\mathbf{S}_{row}$ and $\mathbf{S}_{sym}$, the heatmaps show that $\mathbf{S}_{row}$ retains the block structure better than $\mathbf{S}_{sym}$ and is devoid of the influence of the degree corrections.

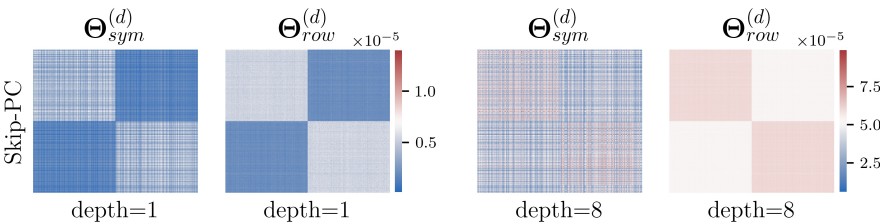

Figure 13: **Numerical validation of DC-SBM for Skip-PC.** It shows the exact NTKs $\boldsymbol{\Theta}^{(d)}$ for $\mathbf{S}_{sym}$ and $\mathbf{S}_{row}$ for depths $d = 1$ and $8$.

In the case of Skip-$\alpha$,we use $\alpha = 0.1$ to obtain the result illustrated in Figure 14. Similar conclusions are derived from the experiment. Although we consider $\mathbf{XX}^T = \mathbf{I}_n$ for Skip-$\alpha$ which fundamentally relies on the feature information to interpolate, the results are still meaningful and demonstrate the theoretical findings.

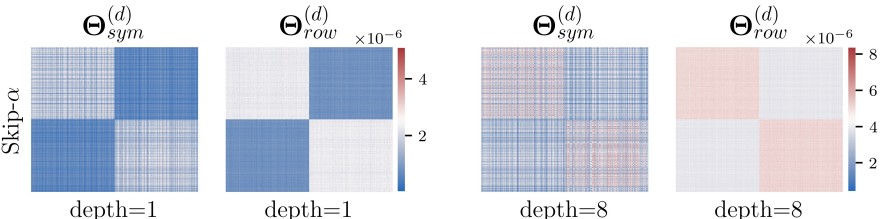

Figure 14: **Numerical validation of DC-SBM for Skip-$\alpha$.** It shows the exact NTKs $\mathbf{\Theta}^{(d)}$ for $\mathbf{S}_{sym}$ and $\mathbf{S}_{row}$ for depths $d = 1$ and 8.

**Numerical analysis of the results using Heterophily DC-SBM.** We extend the analysis to heterophily setting by sampling a graph of size $n = 1000$ and validate our theoretical results on the impact of depth in Vanilla GCN, Skip-PC and Skip-$\alpha$. We plot the NTKs for depth $d = 1$ and $d = 8$ for symmetric and row normalized adjacency matrices and linear GCN for all the cases. Figure 15 illustrates the results for Vanilla GCN where the plot in the first column shows the heterophilic DC-SBM from which the graph is sampled. Observations are similar to the homophilic setting, validating our theoretical results from Theorem 4.

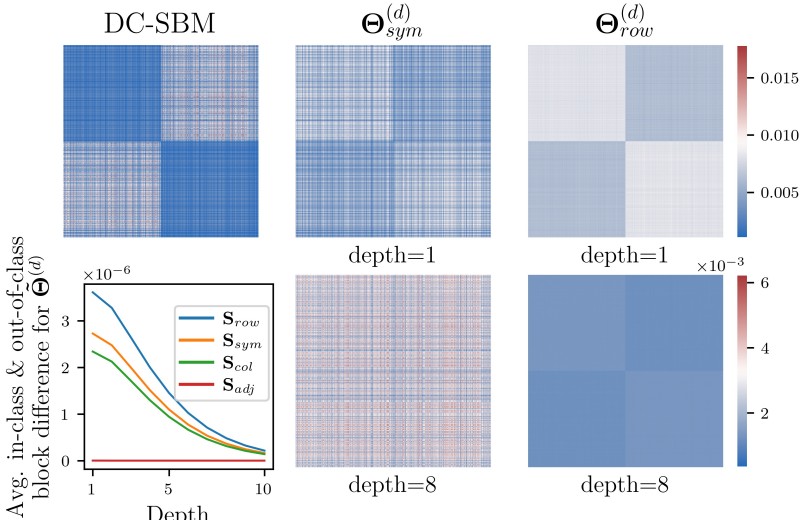

Figure 15: **Numerical validation of Theorem 4 using DC-SBM** shown in the first plot of column 1. Columns 2 and 3 illustrate the exact NTKs of depth=1 and 8 for $\mathbf{S}_{sym}$ and $\mathbf{S}_{row}$, respectively. Second plot in column 1 shows the average gap between in-class and out-of-class blocks from theory.

**Validation of the theoretical filter ordering based on the population kernel block difference**. Similar to the homophily case, we validate the theoretical finding of the filter $\tilde{\mathbf{\Theta}}_{row} \succ \tilde{\mathbf{\Theta}}_{sym} \succ \tilde{\mathbf{\Theta}}_{col} \succ \tilde{\mathbf{\Theta}}_{adj}$ based on the population kernel block difference by sampling a graph from a DC-SBM and measuring the Mean Squared Error (MSE) of the prediction from the exact kernel for various depth of GCN. Figure 16 illustrates the order of convolution filters obtained theoretically holds very well in practice.

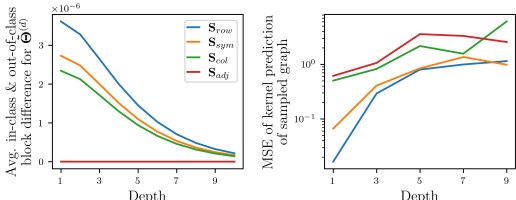

Figure 16: **Numerical validation of the theoretical filter ordering based on the kernel class separability.** Left plot shows the result from theory based on the block difference of the population NTK. The right plot shows the Mean Squared Error (MSE) of the prediction from the exact kernel of a sampled graph. The order of convolutions based on MSE clearly validates the theory.

Figure 17 shows the impact of depth for symmetric and row normalized adjacency in Skip-PC and Skip-$\alpha$ GCNs. Again, we observe similar results as homophilic and also the theoretic results hold such as the class information is still retained for larger depth and the gap doesn't vanish, and between $\mathbf{S}_{row}$ and $\mathbf{S}_{sym}$, the heatmaps show that $\mathbf{S}_{row}$ retains the block structure better than $\mathbf{S}_{sym}$ and is devoid of the influence of the degree corrections.

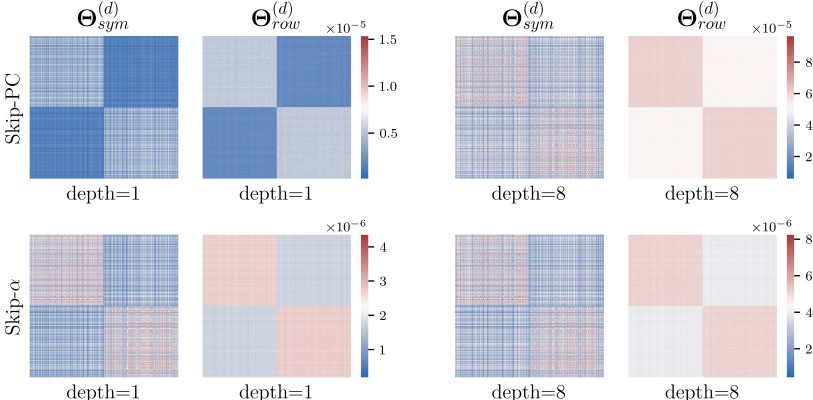

Figure 17: **Numerical validation of DC-SBM for Skip-PC and Skip-$\alpha$.** It shows the exact NTKs $\mathbf{\Theta}^{(d)}$ for $\mathbf{S}_{sym}$ and $\mathbf{S}_{row}$ for depths $d = 1$ and 8.

**Numerical Validation of Core-Periphery DC-SBM**. In this section, we validate the two scenarios discussed in Section 5.3 - core-periphery without community structure and core-periphery with community structure. For the firsr case, we consider core-periphery DC-SBM with $n/4$ nodes as core and the rest as periphery as shown in the first heatmap of Figure 18. We plot the exact NTKs of depth 2 for symmetric and row normalization using Vanilla GCN as shown in the second and third heatmaps of Figure 18. This clearly demonstrates the theoretical result presented in Corollary 2 where the symmetric normalization exhibits the graph structure and the row normalization is a constant kernel.

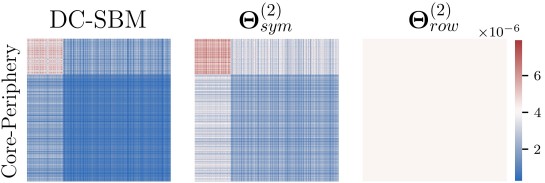

Figure 18: **Numerical validation of Core-Periphery DC-SBM.** It shows the exact NTKs $\mathbf{\Theta}^{(d)}$ for $\mathbf{S}_{sym}$ and $\mathbf{S}_{row}$ for depth 2.

In the second setting, we consider two communities of equal size $n/2$ with core-periphery in each, and the link probabilities between cores of the communities is higher than core-periphery or periphery-periphery of the two communities as shown in the first heatmap of Figure 19. The exact NTKs of symmetric and row normalization are illustrated in the second and third heatmaps of Figure 19 where we see that row normalization retains the community structure again.

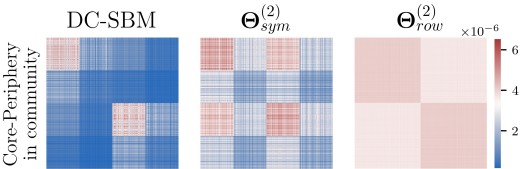

Figure 19: **Numerical validation of Core-Periphery DC-SBM with community structure.** It shows the exact NTKs $\mathbf{\Theta}^{(d)}$ for $\mathbf{S}_{sym}$ and $\mathbf{S}_{row}$ for depth 2.

### C.4 Experiments on Real Dataset: Cora

**Orthonormal Feature $\mathbf{XX}^T = \mathbf{I}_n$ Assumption**. In this section, we present additional experiments on Cora. Since our theory assumed orthonormal features $\mathbf{XX}^T = \mathbf{I}_n$, we validate it experimentally in similar setup described in Section 7. Figure 20 shows the result for $\mathbf{S}_{sym}$ and $\mathbf{S}_{row}$ for depth 1 and 8. The conclusions derived from real setting hold here as well and shows $\mathbf{S}_{row}$ preserves the class information better than $\mathbf{S}_{sym}$.

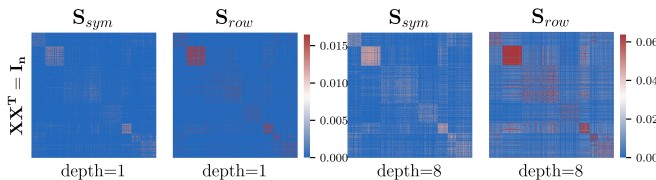

Figure 20: **Evaluation on Cora with $\mathbf{XX}^T = \mathbf{I}_n$.** Plot shows $\mathbf{S}_{sym}$ and $\mathbf{S}_{row}$ for depths $d = 1$ and 8.

**ReLU GCN.** We present the result for ReLU GCN in this section. Figure 21 shows the result where the conclusions derived in Section 7 holds very well. Additionally, we plot the average in-class and out-of-class block difference in the case of vanilla GCN (line plots in first row of Figure 21), we observe that the average in-class and out-of-class block difference degrades with depth for each class in Cora, showing the negative impact of depth which aligns well with the theoretical result.

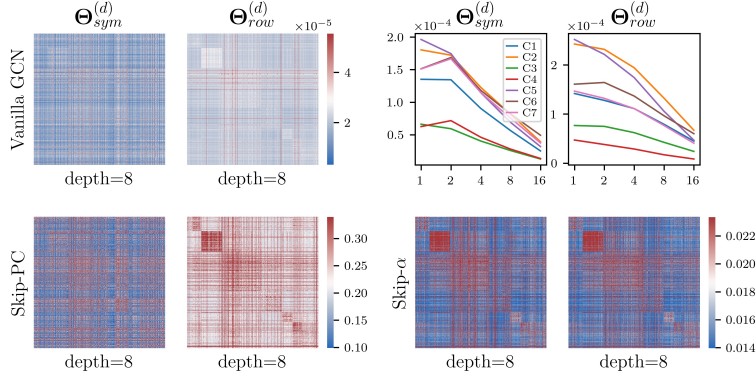

Figure 21: **Evaluation on Cora dataset.** Heatmaps show results of vanilla GCN and the decrease in class separability with depth for $\mathbf{S}_{sym}$ and $\mathbf{S}_{row}$. Last two show NTKs of Skip-PC where a min and max threshold of 30 and 70 percentile is set for better visualization.

Another experimental study is to understand how easy it is to learn the classes that showed good in-class and out-of-class gap preservation from the above experiment. The line plot in Figure 21 shows class $C2$ and $C5$ are well represented by both $\mathbf{S}_{sym}$ and $\mathbf{S}_{row}$. To study how well this holds in the trained GCN, we considered depth 4 vanilla GCN with ReLU activations and used the same hyperparameters mentioned in Section C.1. The results are shown in Figure 22 where we observe that $C2$ and $C5$ are well learnt. On the other hand, other classes that showed small gap are also well learnt by the trained GCN. This needs further investigation as it has to do with the data split and some classes are poorly represented in the training data, for instance $C6$. Thus, we leave it for further analysis.

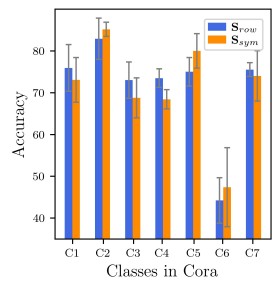

Figure 22: **Class wise performance of trained GCN of depth** 4.

**Linear GCN.** We present the result for linear GCN with the same setup as described in Section 7 to check the goodness of our theory. The results are illustrated in Figure 23 where we observe that the theory holds very well for linear GCN than ReLU GCN. The class information is better preserved in $\mathbf{S}_{row}$ than $\mathbf{S}_{sym}$ especially for higher depth in the case of both GCN with and without skip connections. All the conclusions derived in the main section hold here as well.

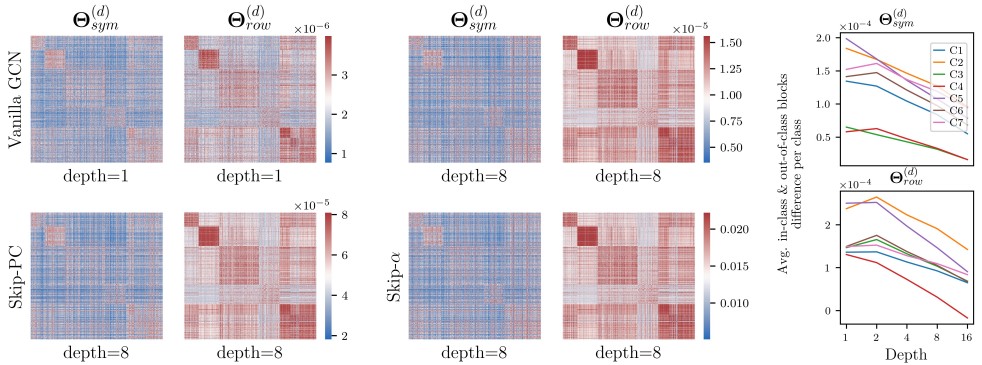

Figure 23: **Evaluation on Cora using linear GCN.** First row shows the results for vanilla GCN for depths 1 and 8. Second row shows the result for Skip-PC and Skip-$\alpha$ for depth 8. The last column shows the average in-class and out-of-class block difference per class of both the symmetric and row normalized adjacencies.

### C.5 Experiments on Real Dataset: Citeseer

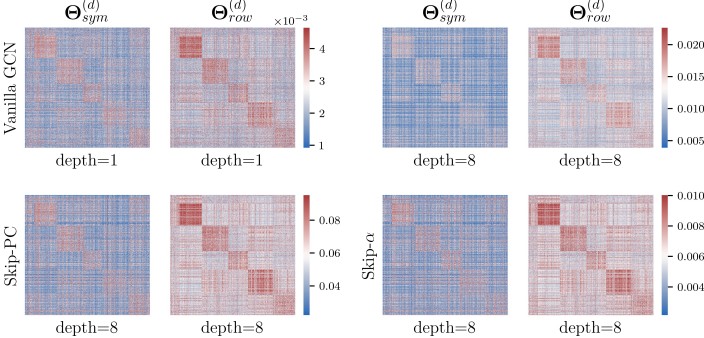

Figure 24: **Evaluation on Citeseer dataset using linear GCN.** First row shows the results for vanilla GCN for depths 1 and 8. Second row shows the result for Skip-PC and Skip-$\alpha$ for depth 8.

In this section, we validate our theoretical findings on Citeseer without much of the assumptions. We consider multi-class node classification ($K = 6$) using GCN with linear activations and relax the orthonormal feature condition, so $\mathbf{X}\mathbf{X}^T \neq \mathbf{I}_n$. The NTKs for vanilla GCN, GCN with Skip-PC and Skip-$\alpha$ for depths $d = 1, 2, 4, 8, 16$ are computed and Figure 24 illustrates the results. All the observations made in Section 7 hold here as well and clear blocks emerge for $\mathbf{S}_{row}$ making it the preferable choice as suggested in the theory.

