# OpenReview forum: "Analysis of Convolutions, Non-linearity and Depth in Graph Neural Networks using Neural Tangent Kernel"
_TMLR — Accepted by TMLR_

### Review · Reviewer_K9MK · 2023-07-20

**Summary Of Contributions:**

1. This paper provides a more general result of NTK of GCN for node classification settings. The analysis covers multi-layer GCNs for multi-community, homophilous and heterophilous graphs.
2. Theoretically and empirically, The paper studies the comparisons between linear GCN and Relu GCN and between different graph convolutions. It also shows an explanation of over-smoothing from the NTK perspective.
3. This work also covers the analysis of skip connections of GCNs using NTK.

**Audience:**

Yes

**Broader Impact Concerns:**

There are no concerns on the ethical implications of the work.

**Claims And Evidence:**

Yes

**Requested Changes:**

I will consider a higher final score if the following can be addressed.

1. Please examine the logic in Section 5 and improve the presentation.

2. The introduction to the random graph model in Section 3 also needs improvement. For example, I don't understand what "degree correction vector" means.

3. Linear GCN is confusing to me at first. Later I realized it refers to SGC. If you want to give it a new name, please clarify it.

4. Some related works of generalization or NTK analysis for GNNs are missing. Please include them in the related works.
[1] Cong et al., 2021, "On Provable Benefits of Depth in Training Graph Convolutional Networks."
[2] Li et al., 2022, "Generalization guarantee of training graph convolutional networks with graph topology sampling."
[3] Tang et al., 2023, "Towards Understanding Generalization of Graph Neural Networks."

**Strengths And Weaknesses:**

Strengths:
1. This analysis is quite novel, from my understanding. The contributions are significant and attractive to the graph neural network community.
2. The analysis seems solid, and the conclusions look correct.
3. The framework is general because it can cover homophilous/heterophilous graphs, graphs with skip connections, and different GCNs.

Weaknesses:
1. Some claims and discussions are not clear or rigorous. For example, in the paragraph on page 7 titled "Comparison of graph convolutions", it is very unclear how you make the comparison by saying whether the kernels are affected by $\pi$ or not. Actually, whenever you want to have some discussions by comparing kernels, I feel the reasoning is not very rigorous, although there are supportive experiments. Another example is right below Figure 4 on page 8; what does it mean by "the population NTKs converge to a constant kernel, **thus** 0 class separability for all the convolutions"? I thought the class separability r should be a given condition for the analysis. Why does it read like a result by "thus 0 class separability"?
2. It is interesting to cover the analysis of heterophily. However, it seems the experiments are only implemented on Cora and Citeseer, which are homophilous graphs. Why not verify the effect of heterophily, even if you use synthetic graphs?

---

> ### Author Response · Authors · 2023-08-19
>
> We thank the reviewer for the constructive feedback. We address the concerns below.
>
> ### Clarification on comparing the kernels using class separability
> We understand that the confusion stems from using ‘class separability’ for both the DC-SBM and kernel. We clarify it in the following by detailing the analysis framework: we analyse the performance of different GCNs by deriving the population NTK for each model and comparing the preservation of class information in the kernel. Note that the true class information in the graph is determined by the blocks of the underlying DC-SBM -- formally by $p$ and $q$ and independent of the degree correction $\mathbf{\pi}$. Consequently, we define the **class separability of the DC-SBM** as $r:=\frac{p-q}{p+q}$.
> Hence, in order to capture the class information, the *kernel should ideally have a block structure that aligns with the one of the DC-SBM*. Therefore, we measure the **class separability of the kernel** as the average difference between in-class and out-of-class blocks of the kernel matrix.
> The best case is indeed when the class separability of the kernel is proportional (due to scale invariance of the kernel) to $p-q$ and independent of $\mathbf{\pi}$.
>
> From the above understanding, we now clarify the specific questions from the reviewer:
> * **Comparison of graph convolutions based on the influence of $\mathbf{\pi}$**
>
> Since the underlying class information is determined solely by the blocks in DC-SBM independent of $\mathbf{\pi}$, the best case is when the resulting population NTK is independent of $\mathbf{\pi}$ and the average in-class and out-of-class block difference (class separability of the kernel) is proportional to the class separability of the DC-SBM ($r:=\frac{p-q}{p+q}$). On this basis, we compare the graph convolutions through their corresponding population NTKs.
>
> * **Clarification on the population NTKs converge to a constant kernel, thus 0 class separability**
>
> Here class separability refers to the class separability of the kernel, which is the average in-class and out-of-class block difference. Since in the limits ($n \to \infty$ and $d \to \infty$), the population NTKs converge to a constant kernel, the average block difference is $0$. Hence, the class separability of the kernel is $0$, and so the kernel has no class information.
>
> We have clarified this in the updated draft by adding a discussion on the analysis framework at the end of Section 3 and clarifying the class separability of kernels in the discussion in Sections 4 and 5. We thank the reviewer for raising this concern.
>
> ###  Analysis of Heterophily graphs
> We would like to highlight that we have included the result for the heterophily case and vanilla GCN in Figure 4 by sampling a graph of size 1000. Additionally, we have updated the draft with Figures 13 and 14 that show the effect of depth and skip connections, Skip-PC and Skip-$\alpha$, for heterophily graphs in the appendix.
>
> ### Updates done to the current version of the paper
> We made the following updates to the latest version of the paper:
> * Improved the introduction of DC-SBM in Section 3. The discussion of the class separability of the kernel and the results are clarified and highlighted now in Sections 3 and 4 - 5, respectively.
> * In the case of heterophily, we have included the discussion of the analysis using a synthetic graph for vanilla GCN, Skip-PC and Skip-$\alpha$ in Figures 13 and 14 in the appendix.
> * Clarified the linear GCN and SGC are the same in Section 2.
> * Added the missing references in Section 8 under related works. Thanks for pointing them out.

---

> > ### Comment · Reviewer_K9MK · 2023-09-10
> >
> > Thank you for your reply. It makes the paper clearer.

---

### Review · Reviewer_X5F1 · 2023-08-14

**Summary Of Contributions:**

The authors use the machinery of the neural tangent kernel to attempt to understand the influence of convolutions based on graph Laplacian with row normalisation. Specifically, they prove that linear networks capture class information as well as ReLU,row normalisation preserves underlying class structure better than other convolution choices, performance decreases with network depth and skip connections retain class information even for infinitely deep models.

**Audience:**

Yes

**Broader Impact Concerns:**

I do not have any immediate broader impact concerns.

**Claims And Evidence:**

Yes

**Requested Changes:**

Please address the weaknesses I mentioned above.

**Strengths And Weaknesses:**

Strengths:
- The paper appears to cite a good amount of relevant literature.
- The problem and content of this paper is likely to be of interest to the TMLR community.
- The results and flow of the paper are mostly well presented, and the theorem statements are relatively clear. I did not check the proofs in the appendix.

Weaknesses:
- Precision can generally be improved. For example, the notation $\dot{\sigma}$ is used for the derivative in the notation section. However, the ReLU activation, which does not admit a classical derivative, is mentioned earlier and throughout the text. Then, in Theorem 1, a statement is made iinvolving $\dot{\sigma}$. It is not clear whether the authors intend to claim that this theorem applies to the ReLU, since no qualification on $\sigma$ is provided in the theorem conditions. If the authors are claiming that this theorem applies, how should one understand $\dot{\sigma}$?
- Assumption 1 seems very strong to me. If $X \in \mathbb{R}^{n \times f}$, does $X X^\top = I$ require that $f \geq n$, i.e. the dimension of the node feature is greater than the number of features? Even under this requirement, one still then requires the features to be orthogonal, which seems implausible and also not expressive.

---

> ### Author Response · Authors · 2023-08-19
>
> We thank the reviewer for the positive review, and we clarify the concerns in the following.
>
> ### Derivative of $\sigma$ in Theorem 1
> In general, Theorem 1 is derived without any assumption on $\sigma$. For any $\sigma$ where $\dot{\sigma}$ exists, this expression can be used directly. However, even in cases of functions that are not everywhere differentiable (such as ReLU), an analysis is possible (under slight abuse of notation of  $\dot{\sigma}$). In the case of ReLU, a rigorous analysis of the considered quantities is provided in [1] by exploiting the homogeneity of the function to derive the expectations in the NTK expression. We state the result in Corollary 5 in Appendix A.6 for ReLU and derive the population NTKs for the normalized graph convolution.
>
>
> ### Clarification on Assumption 1
> The orthonormal features assumption is made only to simplify the NTK expressions. This is done to focus only on the influence of the different convolution operators, since $X$ has the same contribution for all GCNs.  In Appendix A.8, we have provided the analysis without the orthogonality assumption but instead assuming a Contextual Stochastic Block Model (where features also have class information). This eads to the same conclusions about the filters. We have clarified this in the ‘Remark on Assumption 1’ paragraph in Page 4.
>
> [1] Arora, S., Du, S.S., Hu, W., Li, Z., Salakhutdinov, R.R. and Wang, R., 2019. On exact computation with an infinitely wide neural net. Advances in neural information processing systems, 32.

---

> > ### Comment · Reviewer_X5F1 · 2023-09-26
> >
> > Thanks for clarifying about the derivative assumption and the orthonormal features assumption. I do not have significant concerns regarding correctness, and I think some TMLR readers will find this paper interesting. I'm glad that the authors added some extra experimental figures as well in response to Reviewer A2PN's comments, however I do not that Reviewer A2PN's concerns are valid. Perhaps the authors could consider reducing the strength of their claim on their experimental results. Overall, I recommend acceptance.

---

> > > ### Author Response · Authors · 2023-09-28
> > > **Thank you**
> > >
> > > We thank the reviewer for recommending acceptance of our paper. We will tone down the experimental claim wherever necessary in the final version.

---

### Review · Reviewer_A2PN · 2023-08-31

**Summary Of Contributions:**

The paper analyzes deep ReLU or linear GNNs in the Neural Tangent Kernel (NTK) regime under the Degree Corrected Stochastic Block Model (DC-SBM). The authors measure class separability of the NTK (mean absolute difference between in- and out-of-class entries) which is suggested to be indicative of the respective GNN, and establish that

* Linear GNTK has class separability equal or better to those of the ReLU GNTK;

* Row normalization aggregation has better class separability than other alternatives;

* Class separability degrades slowest with depth for row normalized aggregation GNTK (yet degrades to random chance at infinite depth).

* Skip connections allow GNTK to not degenerate at infinite depth and preserve class separability.

The authors perform some preliminary experiments to establish the claims on two real datasets, suggesting that presented theory could explain why in prior works linear GNNs have performed on par with ReLU GNNs, and why row normalization sometimes empirically outperforms symmetric normalization.

**Audience:**

Yes

**Broader Impact Concerns:**

No concerns.

**Claims And Evidence:**

No

**Requested Changes:**

* To demonstrate best agreement with theory, I believe Figure 1 and 2 should include results for depth $d = 1$. It is common for any dataset and architecture combination to have a depth sweet spot where accuracy falls off monotonically to both sides of the ideal depth (see for example https://arxiv.org/pdf/2010.09610.pdf), and I wonder if $d = 2$ could just happen to be the best accuracy for *Cora* for reasons unrelated to the theory.

    * Highly-optionally: results would look even more robust if full range of depth up to 8 were evaluated.

* Moreover, could you extend your analysis and all experiments to depth $d=0$ NTK / GCN? My potential concern is that at $d = 0$ class separability of the kernel might be good, while empirical performance of such shallow GCNs may be lacking, which also risks invalidating the theory.

* I wish all plots demonstrating kernel class separability were accompanied by the actual performance (accuracy, mean squared error) of the respective exact / population NTK(s) (and, optionally, even GNNs) on the graph(s). In addition or alternatively, I wish it was spelled out in more detail / more formally why the kernel class separability as defined in section 3 is the metric most predictive of performance (i.e. why look at the in-out class entries absolute difference and not say their ratios, squared distance or some other properties of these matrices).

	* Further, could class separability of the kernel be measured on the Cora dataset, to plot along Figures 1 and 2 to further validate the theory?

* Page 10, "Infact" -> "In fact"

**Strengths And Weaknesses:**

Disclaimer: I am not an expert on literature and did not verify the mathematical details.

### Strengths:

* The idea of analysis of nonlinearity and aggregation mechanism using population NTK under DC-SBM appears novel and well-executed. The theoretical results appear robust.

* The paper is overall well-written and clear.


### Weaknesses:

* The presentation does not fully convince me that presented theory indeed explains empirical performance of GNNs. There is only one experiment (Figure 1/2) measuring actual accuracy of GNNs and the trends on it aren't robust enough (large error bars) or don't always match theory predictions. For example:

	* Section 4 predicts ReLU to do more over-smoothing at large depth than linear networks, but Figure 1/2 shows the opposite trend for row normalized networks.

    * Figure 3 predicts that row normalization will outperform symmetric most at shallow depths, but Figure 1/2 doesn't really show such a trend for linear networks (and the ReLU trend is inverted).

    * Section 6 predicts better performance of skip connections at large depth, but it's not visible on Figure 7 (leftmost vs rightmost heatmaps; in general, it's hard to analyze this Figure visually).

    * Minor: one of the conclusions of section 5.1 is that depth is detrimental to GCNs. Degenerating NTK at infinite depth is a common observation for most architectures, but it isn't actionable for depth selection and unlikely to explain the downward trend in Figure 1/2, as these depths are still too small (see more on this in Requested Changes).


	* See more actionable concerns in Requested Changes.


On the whole I find the paper to study a very interesting theoretical setting with robust theoretical findings, but I think that it lacks actual experiments (GNN and GNTK accuracy, real-world and toy) to convince me that this setting is relevant to real-world GNN tasks. As such, I find this work to be of interest to the TMLR audience, but the claims and evidence are only convincing within the specific theoretical setting; extensions to real GNNs / datasets aren't very compelling.

---

> ### Author Response · Authors · 2023-09-14
>
> We thank the reviewer for the positive review, and we address the requested changes by adding new plots that can be found in the revised version of the paper.
>
> ### Experimental results for more depths in Figures 1 and 2
> We added a new figure to the appendix (Figure 8) showing the results for depth=0 and 1. Note that depth in our GCN definition (equation 2) refers to the number of hidden layers and so depth=0 refers to no hidden layers in the GCN. We added the figure to the appendix for better readability, as the performance for $d=0$ is lower, making the degradation of values for higher depth hard to distinguish. Indeed we observe that the accuracy for $d=0$ is significantly lower than for higher depths as expected. Furthermore, this observation is consistent with the analysis of Cora in [1] (Figure 5), where the performance of GCN for a full range of depth till $10$ is presented.
>
> ### Analysis for $d=0$
> As defined in Section 2, the NTK analysis is done by considering the hidden layers to be in the infinite width setting. Hence, we allowed for at least one hidden layer in the definition of GCN ($d \geq 1$ in equation 2). In the setting of $d=0$, the GCN would reduce to $\sigma(SXW)$, implying the dimension of the weight matrix is $f \times K$ and therefore, a limiting analysis is not possible (The kernel would not stay constant during training). While $\sigma(SXW)$ might be an interesting quantity to study in its own right, there does not exist any NTK equivalent and is therefore not covered by our analysis.
>
> ### Relation between Kernel Class Separability and GCN Performance
> We added the performance evaluation of homophily and heterophily cases in Figures 12 and 16 in the appendix, respectively, where we plot the class separability of the population NTK on the left and the Mean Squared Error of the prediction of the exact NTK on the right. These plots illustrate clearly that class separability is a good metric to study the performance of the GCN as it models the observed ordering of convolution operators perfectly. We clarified the idea of using class separability in the response to reviewer K9MK as well as added a discussion of the class separability of the kernel, and clarified the results in Sections 3 and 4 - 5, respectively, in the current draft. In general, we acknowledge that the metric chosen in this paper is not a unique proxy for the performance of the network, however, as observed in the numerical evaluation, the conclusions from the class separability of the kernel translate well to the observed performance of the GCN.
>
> ### Class Separability of the Kernel and Performance on the Cora Dataset
> We have already presented this analysis in the submitted version as part of B.4 in the Appendix (refer to Figures 21 and 22 along with the corresponding discussions). We showed the class separability per class for Cora dataset for ReLU GCN in the line plots of Figure 21, where the overall in/out of class difference decreases with depth, being in line with the accuracy shown in Figures 1 and 2. Furthermore, we extended the analysis to study how well the ordering of the in/out of class difference per class translates to the actual class wise performance of the trained GCN in Figure 22 (for depth 4). While we observed that the classes with high kernel in/out class separation (C2 and C5) are learned with high accuracy, the other classes with small kernel separation are also reasonably learned. This is mainly due to the poor representation of some classes in the dataset and we leave it for future analysis. We added the kernel class separation per class for linear GCN in the last column of Figure 23.
>
> ### Updates done in the revised version
> Added Figures 8, 12, 16 and 23 with discussions in the Appendix to address the comments.
>
> We thank the reviewer for pointing the concerns in the gap between theory and observations on real data, which is a general problem for deep learning. If the reviewer can suggest further studies, we can try to revise accordingly.
>
> [1] Thomas N. Kipf and Max Welling. Semi-supervised classification with graph convolutional networks. In International Conference on Learning Representations (ICLR), 2017.

---

> > ### Comment · Reviewer_A2PN · 2023-10-03
> > **Thank you for clarifications and updates**
> >
> > Thank you for the detailed replies. Updated results (including $d=1$; showing some correlation between block difference and MSE; pointing out the Cora class separability measurements) are on the whole encouraging, so I am leaning for a weak accept.
> >
> > Parting comments:
> > * I recommend propagating the discussed issues/updates much more into the main text;
> > * There is inconsistency in using MSE vs Accuracy in Figures (e.g. Figure 1/22 vs 12/16); it would be best to show both metrics everywhere.
> > * Figures 12/16 still show that there is a big difference between block difference and MSE, e.g. at depth 5 row-normalized NTK has much lower block difference than other architectures ad depth 1, but its MSE is still much lower. So clearly class separability is far from a perfect accuracy/loss proxy, and hence deserves even more justification in the text and caution when drawing conclusions, perhaps some analysis on where it works and where it breaks etc.
> > * Furthermore, regarding $d=0$, from my understanding your current results could be low due to underfitting since a $d=0$-GCN has too few parameters. But we could consider a network with $F_W = X W_1 W_2$ and/or $F_W = S X W_1 W_2$, optionally with or without nonlinearities. These could be sufficiently overparameterized via a large hidden layer, while also having only $0$ or $1$ application of $S$ (IIUC you now have at least 2 convolutions in the $d=1$ setting). I’m curious how class separability / accuracy / mse would behave then, and whether this would also be a setting where they don’t correlate (similarly to the above setting of comparing GCNs of different depths using different normalizations).

---

> > > ### Author Response · Authors · 2023-10-06
> > >
> > > We agree with the reviewer that the kernel class block difference does not exactly capture the MSE or accuracy of the network. However, it does capture the overall trends as shown in the experiments. That being said, we have toned down the experimental claim wherever necessary in the updated version (also mentioned in our response to Reviewer X5F1).
> > >
> > > **MSE vs Accuracy:** we chose to plot MSE as the considered block model has good separation ($p=0.8$ and $q=0.1$ for homophily, and vice versa for heterophily as stated in the experimental details in Section B.3) and hence perfect classification for shallow depths.
> > >
> > > Regarding $d=0$, one could consider redefining the network as suggested by the reviewer. However, it is a slightly different network compared to our definition in (2). To analyze the new setup, we considered $S^kXW_1W_2$ as a GNN of depth $1$ with $k$ convolutions. We vary $k$ between $0$ and $10$ for a homophily graph sampled from DCSBM as described in B.3. We plot the kernel block difference and MSE in the figure titled 'tmlr_sgc_k.jpg’ uploaded to the supplementary files. Note that the plot shows $k=0$ performs poorly in both the class separation and MSE indicating that at least one convolution is needed. The class separation captures the overall trend in MSE, but not every depth exactly. We did not want to add the plot to the draft as it is a different architecture and therefore does not align with the main analysis provided in the paper.

---

### Author Response · Authors · 2023-11-06
**Latest update to the paper**

We found a small technical error in the derivation of the neural tangent kernel for graph convolutional networks (Theorem 1). We fixed it with equation (7) and derived the NTK. Consequently, we reevaluated the following,
* We computed the population NTKs for linear GCN (Theorem 2), ReLU GCN (Theorem 3), and all four convolutions (Theorem 4).
* The class separability measure of the kernels that we use to analyze is included in Theorem 2 as we could compute the exact expressions with the corrected NTK.
* We changed Corollaries 1, 5, and 6 to class separability of the population NTK, instead of the population NTK itself, as an explicit expression is possible to compute now and our results are based on this measure.
* We updated all the figures that involved NTKs such as Figures 2–7 in the main draft and Figures 10–21, 23, and 24 in the Appendix. We also updated the discussion wherever necessary. However, the main message and the theoretical implications of the paper remain unchanged.

---

### Decision · Action_Editors · 2023-10-06

**Recommendation:** Accept as is

**Comment:**

The paper cites a good amount of relevant literature, but I think there is still room for improvement. A novel contribution of the paper is to apply signal propagation (propagation of NTK from layer to layer) to analyze properties (e.g., expressivity) of graphical neural networks. There is a line of work, dating back to 2016, that uses this technique (signal propagation) to analyze expressivity, spectrum, and trainability of neural networks. A partial list of them includes: 1. MLP (https://arxiv.org/abs/1606.05340, https://arxiv.org/abs/1611.01232), 2. residual networks (https://arxiv.org/abs/1712.08969), 3. CNN (https://arxiv.org/abs/1806.05393), 4. RNN & LSTM (https://arxiv.org/abs/1806.05394, https://arxiv.org/abs/1901.08987). 5 general deep networks (https://arxiv.org/abs/1912.13053), 6. Tensor programs  (https://arxiv.org/abs/1910.12478, cited in the paper). It would be good to discuss the connection between the current work and the above work, bridging the two communities.

**Audience:**

Yes. Both graphical neural networks and the theory of infinite-width neural networks are important and active research topics in machine learning and deep learning.

**Claims And Evidence:**

The paper uses the machinery from infinite-width networks, i.e., neural tangents kernels, to analyze the influence of several design choices (activation, row-normalization, skip-connection, etc.) in GNN. The main technical contribution is to derive the recursive formula of the NTKs and use this formula to analyze signal propagation in the networks. All reviewers are in agreement that the results from the paper are well-presented and interesting to the community. In addition, the theoretical results are robust [A2PN] and novel [K9MK]. They all recommend acceptance of the paper (2 leaning accept + 1 accept.)